# The adeno-associated virus Rep proteins target PP4:SMEK1 by preventing substrate recruitment

Bram Vandewinkel[1], Sophie Torrekens[1], Zander Claes[2], Mathieu Bollen[2], Els Henckaerts[1,3]*

**1** Trellis Research Group, Department of Cellular and Molecular Medicine, KU Leuven, Leuven, Belgium, **2** Laboratory of Biosignaling & Therapeutics, Department of Cellular and Molecular Medicine, KU Leuven, Leuven, Belgium, **3** Virus-Host Interactions & Therapeutic Approaches Research Group, Department of Microbiology, Immunology and Transplantation, KU Leuven, Leuven, Belgium

* Els.henckaerts@kuleuven.be

## Abstract

Despite the widespread use of adeno-associated virus (AAV) vectors in gene therapy, their clinical efficacy and large-scale manufacturing remain constrained by an incomplete understanding of the virus-host interactions that govern AAV gene expression and replication. Here, we identify the PP4:SMEK1/2 phosphatase complex as an important regulator of wild-type AAV replication. Binding studies show that the AAV replication proteins engage SMEK1 to negatively influence PP4 activity. Specifically, AAV Rep68 interferes with substrate recruitment to the PP4:SMEK1 complex, resulting in hyperphosphorylation of the PP4 substrates KAP1$^{S824}$ and RPA2$^{S4/8/33}$, which in turn enhances viral gene expression and replication. We further uncover a direct interaction between KAP1 and SMEK1, mediated by a MAPP short linear motif that binds the SMEK1 EVH1 domain. Additionally, we identify a multifunctional complex comprising PP4:SMEK1 and PP1:NIPP1 that contributes to KAP1$^{S824}$ dephosphorylation. These findings reveal a previously unrecognized mechanism by which viruses subvert host phosphatases to promote replication. This mechanistic insight not only advances our understanding of AAV and phosphatase biology but also has the potential to inform strategies for enhancing AAV vector potency.

## Author summary

Adeno-associated virus (AAV) is a small, harmless virus with a simple single-stranded DNA genome. Its four replication (Rep) proteins act as master regulators of the viral life cycle by taking control of key cellular machinery. Yet, many important interactions between AAV Rep proteins and host cell factors are still not well understood. In this study, we identify the PP4:SMEK1 phosphatase complex as a previously unrecognized obstacle to AAV gene expression and replication. We show that AAV overcomes this barrier by using its Rep proteins

**Data availability statement:** The source data will be deposited at Figshare (https://doi.org/10.6084/m9.figshare.c.7911407).

**Funding:** This work was supported by the Research Foundation Flanders (G0C3220N; 1S58121N to BV) and the KU Leuven Research Fund (3M190475). The funders had no role in study design, data collection and analysis, decision to publish, or preparation of the manuscript.

**Competing interests:** The authors have declared that no competing interests exist.

to block PP4 activity, resulting in the hyperphosphorylation of host proteins KAP1 and RPA2, necessary for enhanced AAV gene expression and replication. Our findings also reveal that PP4:SMEK1 and PP1:NIPP1 form a large, multifunctional complex that works together to dephosphorylate KAP1. These insights deepen our understanding of how AAV interacts with host cells and may help improve the design and manufacturing of AAV-based gene therapy vectors.

## Introduction

Protein phosphatases are essential regulators of cellular signaling, serving to balance the activity of protein kinases. Once considered mere housekeeping components, they are now recognized as integral regulators of diverse cellular pathways and have been implicated in the pathogenesis of numerous diseases [1]. Given their role in cellular processes such as cell cycle regulation, transcription, protein synthesis and apoptosis, it is not surprising that viruses have evolved mechanisms to hijack cellular phosphatases to benefit their replication. A growing body of research highlights the critical role of reversible protein phosphorylation in the replication of viruses, and both phosphorylation and dephosphorylation of viral and cellular proteins have been shown to play key roles in supporting the life cycle of many viruses. Viruses manipulate serine/threonine phosphatases like PP1 and PP2A to alter host cell processes—either inhibiting or activating them to promote replication, suppress immune responses, or induce cell cycle changes. For instance, HIV and SV40 inhibit PP2A to arrest cells [2], while EBOV uses it to activate transcription [3]. PP1 is similarly hijacked to boost viral transcription and evade host defenses [4]. Moreover, our previous studies on the interactions between the adeno-associated virus (AAV) and PP1 revealed that viruses also manipulate cellular phosphatases to counteract KAP1-mediated chromatinization and silencing of the viral genome [5].

AAV is a non-pathogenic parvovirus with a 4.7 kb ssDNA viral genome, containing two major open reading frames (ORFs) [6,7]. In total, nine different viral proteins are expressed during active replication [7–9], a process that requires the presence of a helper virus such as adenovirus or herpes simplex virus [10,11]. Establishment of the lytic phase of the viral life cycle depends on the expression of various helper virus proteins and the AAV non-structural replication (Rep) proteins. These Rep proteins play an essential role in viral transcription, replication and packaging [12–15] and are expressed from the same ORF in the AAV genome through a complex interplay of alternative promoter usage and splicing events. The four AAV Rep isoforms—Rep78, Rep68, Rep52, and Rep40—share an AAA⁺-ATPase/helicase domain. Rep78 and Rep68 also have a N-terminal endonuclease domain, but only Rep78 and Rep52 have a C-terminal zinc finger. To extend their function beyond what is provided by their individual domains, these proteins assemble into various oligomeric states [16,17] and interact with host cellular proteins to optimally exploit and control the host cell [18].

Various protein-protein interaction approaches have been employed to reveal cellular interaction partners of the Rep proteins, however the molecular mechanisms

and purpose of these interactions remain largely unexplored. Using an unbiased BioID approach to determine the putative Rep interactome, we discovered that the viral replication proteins interact with the PP1:NIPP1 holoenzyme and inhibit its activity [5]. This inhibition maintains the transcriptional co-repressor KAP1[S824] in a phosphorylated, inactive state, causing the dissociation of histone-modifying proteins (complexes) from KAP1, thereby facilitating viral transcription. Notably, dephosphorylation of KAP1[S824] has also been reported to be regulated by protein phosphatase 4 (PP4) [19], a member of the same phosphatase family as PP1. Interestingly, the PP4 regulatory subunit SMEK1 (PP4R3A) was also identified as a putative Rep interactor [5].

PP4 belongs to the phosphoprotein phosphatase (PPP) family and is considered a PP2A-like phosphatase. Although the regulation of its activity, substrate specificity, and cellular localization appear less complex than those of PP1 and PP2A, PP4 remains relatively understudied. The PP4 catalytic subunit forms both heterodimeric complexes (with PP4R1 or PP4R4) and heterotrimeric complexes (with PP4R2 and either PP4R3A or PP4R3B). Among these, the heterotrimeric complexes are currently the best characterized. The regulatory subunits PP4R3A and PP4R3B, also known as SMEK1 and SMEK2, respectively, recruit FxxP or MxPP short linear motif (SLiM)-containing substrates via their N-terminal EVH1 domains for dephosphorylation by associated PP4 [20]. Only a limited number of validated PP4:SMEK1/2 substrates have been identified, including KAP1, RPA2 and γH2AX, suggesting that PP4 primarily functions in processes such as the DNA-damage response (DDR) and cell cycle regulation [19,21,22]. Furthermore, the role of PP4 holoenzymes in the viral life cycle is largely unexplored, and to date, no viral interactions with PP4:SMEK complexes have been reported.

Here, we report for the first time that the PP4 phosphatase, in complex with its regulatory subunit SMEK1, is targeted by viral proteins. We demonstrate that PP4, together with SMEK1 and SMEK2, plays a repressive role in the AAV life cycle. To overcome this restriction, the viral Rep proteins interfere with PP4:SMEK activity, maintaining substrates such as KAP1 and RPA2 in a hyperphosphorylated state during lytic replication. We further show that AAV Rep proteins employ a novel, previously undescribed mechanism of PP4 substrate-recruitment interference. Finally, we identify a large multifunctional phosphatase complex composed of the PP1:NIPP1 and PP4:SMEK1 holoenzymes, suggesting that AAV Rep proteins simultaneously target multiple phosphatase pathways.

## Results

### AAV Rep proteins interact directly with SMEK1

To confirm the data gathered from our previously performed Bio-ID analyses [5], which indicated a potential interaction between the AAV Rep proteins and the PP4:SMEK1 holoenzyme complex, we performed GFP-trap experiments with GFP-tagged SMEK1 from cells that were co-infected with AAV2 (10 IU) and Ad5 (MOI 5) (Fig 1A). Pull-down of GFP-SMEK1 at 28h post-infection (pi) without formaldehyde crosslinking resulted in robust co-precipitation of the PP4 catalytic subunit, all four Rep isoforms, as well as the PP4 substrate KAP1 [19], and SF3B1, a PP1:NIPP1 substrate not previously reported to associate with the PP4 holoenzyme (Fig 1B) [23]. Formaldehyde crosslinking was employed because prior attempts to detect the interaction between Rep and the PP1:NIPP1 holoenzyme had proven challenging [5]. The enzyme-substrate interaction between SMEK1 and KAP1 seems to be enhanced by prior crosslinking, whereas the SMEK1:Rep interaction diminished upon crosslinking (Figs 1B and S1A). Proximity-ligation assays (PLA) on non- and co-infected cells with AAV2 (1000 IU) and Ad5 (MOI 5), as well as reciprocal pull-down assays with overexpression of the four different Rep isoforms confirmed the SMEK1:Rep and SMEK1:KAP1 interactions (S1B and S1C Fig).

To get more insight into the interaction dynamics, we next performed lysate-based split-luciferase assays. This assay, employed to study protein-protein interactions, relies on the separate fusion of two catalytically inactive NanoLuc fragments (LgBiT and SmBiT) to two proteins of interest. The lysate-based variant was recently developed and successfully used to study the interaction of various PP1 and PP2A holoenzymes [24–27]. We fused SMEK1 to the LgBiT tag, while Rep68 and KAP1 were tagged with SmBiT, with KAP1 serving as a positive control (Fig 1C). Rep68 was selected as a representative for studying interactions between the Rep proteins and SMEK1, due to its ease of purification from

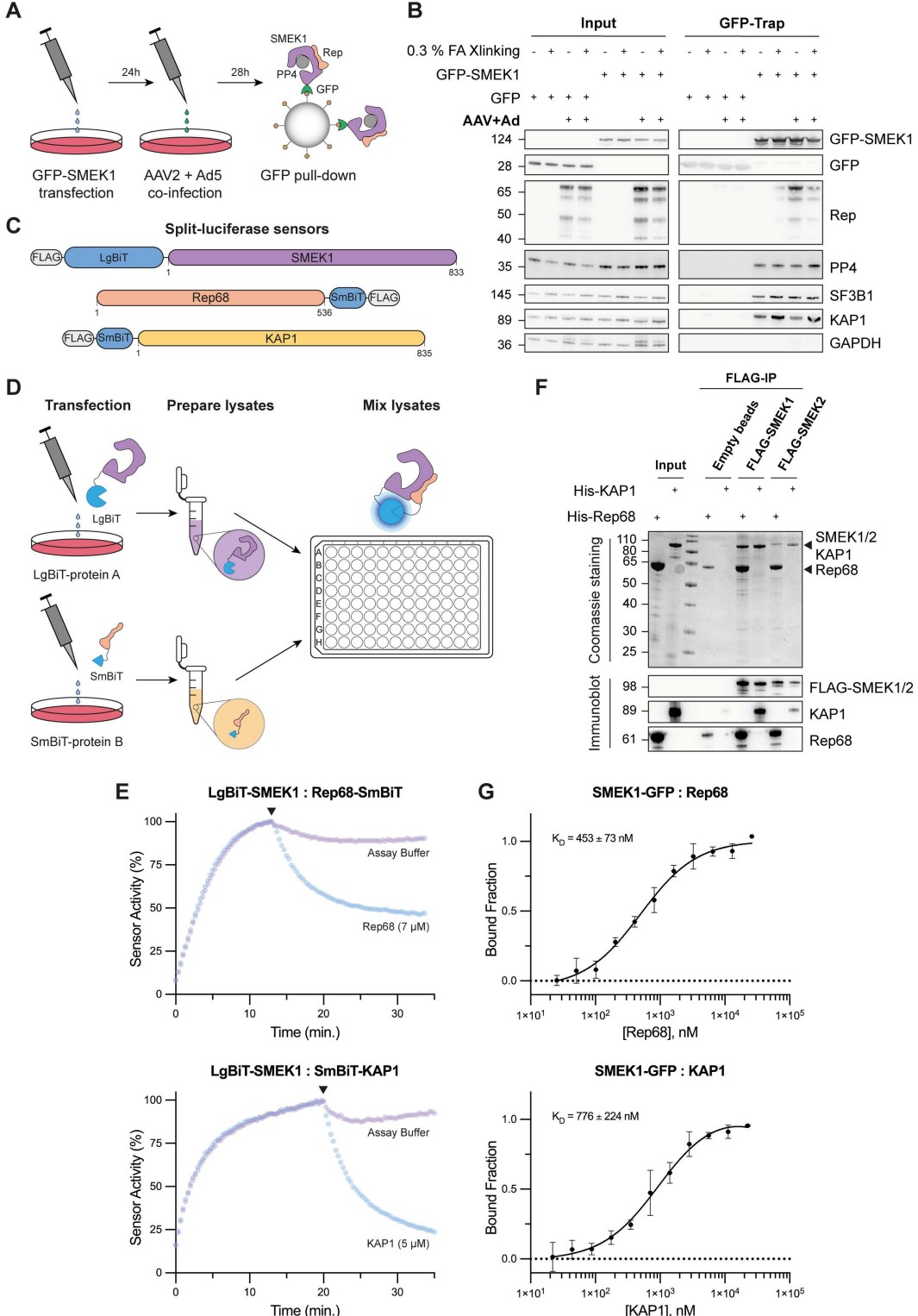

**Fig 1. PP4R3A (SMEK1) directly interacts with the AAV Rep proteins. (A)** Schematic representation of the GFP-trap experiment of GFP-tagged SMEK1 during a lytic AAV infection. **(B)** Co-immunoprecipitation of Rep, PP4, SF3B1 and KAP1 with ectopically expressed GFP-SMEK1, in the presence/absence of formaldehyde crosslinking. GFP alone is included as a control to account for non-specific interactions. Input samples are shown on the left, while pull-down/trap samples are shown on the right. AAV Rep expression was visualized using a mouse monoclonal anti-Rep antibody

(Progen; 61069) recognizing all four Rep isoforms. **(C)** Design of the split-luciferase sensors used for protein-protein interaction studies. SMEK1 was N-terminally tagged with the LgBiT tag, while Rep68 and KAP1 were respectively C- or N-terminally tagged with the SmBiT tag. A flexible linker of 10-15 AA was included in between the tag and protein. **(D)** Experimental design of the lysate-based split-luciferase assays performed with the LgBiT-SMEK1:Rep68-SmBiT and LgBiT-SMEK1:SmBiT-KAP1 sensors as described before [27]. Assays were performed in a white 384-well low protein binding plate. **(E)** Kinetic-trace experiment showing the time-dependent association of the LgBiT-SMEK1:Rep68-SmBiT and LgBiT-SMEK1:SmBiT-KAP1 split-luciferase sensors. The black arrow indicates the addition of the purified, untagged competitor (concentration indicated in the graph). The represented data is plotted as a percentage of the signal-to-background (S/B) ratio right before the addition of the competitor. The data shown is a representative example of three independent repeats. **(F)** FLAG-IP of purified FLAG-SMEK1 and FLAG-SMEK2 transiently expressed in HEK293T cells. Purified FLAG-SMEK1/2 bound to anti-FLAG agarose beads was incubated with purified His-tagged KAP1 or Rep68 to investigate co-precipitation. Co-immunoprecipitation of Rep68 and KAP1 was confirmed via Coomassie staining (top panel) and immunoblotting (bottom panel). Empty beads (beads incubated with non-transfected HEK293T lysate) served as a control to assess the stickiness of the purified Rep68 and KAP1. **(G)** MST assays for the SMEK1-GFP:Rep68 and SMEK1-GFP:KAP1 interaction. The average $K_D \pm$ SD of three independent repeats is indicated in both graphs.

*E. coli* with high yield. To study the dynamic behaviour of the SMEK1:Rep68 and SMEK1:KAP1 interaction, we prepared HEK293T cell lysates from cells that separately expressed LgBiT-SMEK1, Rep68-SmBiT and SmBiT-KAP1 (Fig 1D). To monitor the association of each interaction sensor, lysates containing LgBiT-SMEK1 were combined with either Rep68-SmBiT or SmBiT-KAP1, and luminescence was measured over time. To assess the specificity and dynamic behaviour of these interactions, we introduced purified His-tagged partner proteins lacking the SmBiT tag (i.e., His-Rep68 or His-KAP1, hereafter referred to as "untagged competitor") as competitors (S2A and S2B Fig). The untagged competitors were added at relatively high concentrations (low micromolar) to effectively outcompete the SmBiT-tagged variants, leading to sensor dissociation (Fig 1D, 1E). Upon addition of untagged competitor, dissociation was observed for both interaction sensors (Fig 1E), strongly indicating that the interactions detected in the lysate-based split-luciferase assays are specific and dynamic.

We extended the lysate-based assays with an *in vitro* pull-down approach to study the direct interaction of purified Rep68/KAP1 and SMEK1. Ectopically expressed FLAG-SMEK1 was purified from HEK293T cells. Next, the purified FLAG-SMEK1 bound to the beads was incubated with either recombinant Rep68 or KAP1. Following two washing steps, Rep68 and KAP1 co-precipitated with purified FLAG-SMEK1, suggesting a direct interaction with SMEK1 (Figs 1F and S2D). We also included FLAG-SMEK2-bound beads and observed that Rep68 also interacts strongly with the alternative PP4 regulatory subunit that constitutes the other heterotrimeric PP4 complex (Fig 1F). To further validate the direct interaction with SMEK1, we performed split-luciferase assays using recombinantly expressed and purified LgBiT-SMEK1, Rep68-SmBiT, and SmBiT-KAP1 (S2A-S2C Fig). Consistent with previous results, the purified interaction sensors showed clear association, which was reversed by addition of the untagged competitor (S2C Fig). Finally, microscale thermophoresis (MST) revealed that the dissociation constants ($K_D$) of the SMEK1:Rep68 ($K_D = 453 \pm 73$ nM) and SMEK1:KAP1 ($K_D = 776 \pm 224$ nM) interactions were both in the high nanomolar range, with a slightly higher affinity for the SMEK1:Rep68 interaction (Figs 1G and S2E).

In summary, we demonstrated an interaction between the AAV Rep proteins and the PP4 regulatory subunit SMEK1. In addition, *in vitro* assays with purified proteins revealed that SMEK1 interacts directly with Rep68 and KAP1 with high nanomolar affinities.

## Rep68 interacts with the HEAT/Arm domain of SMEK1, independent of PP4

To gain more insights in how PP4 and the Rep proteins bind to SMEK1, we made deletion mutants of SMEK1 (Figs 2A and S3A). The N-terminal EVH1 domain of SMEK1 plays a critical role in substrate recruitment by recognizing short linear motifs (SLiMs), specifically those containing FxxP or MxPP sequences [20]. The HEAT/Arm domain functions as a scaffold for coordinating PP4 assembly [28], while its unstructured C-terminal region contains its nuclear localization signal (NLS) [29]. SMEK1 truncations were tested for loss of PP4 binding using lysate-based split-luciferase assays. As expected, deletion of the HEAT/Arm domain of SMEK1 resulted in a complete loss of binding with PP4, whereas removal

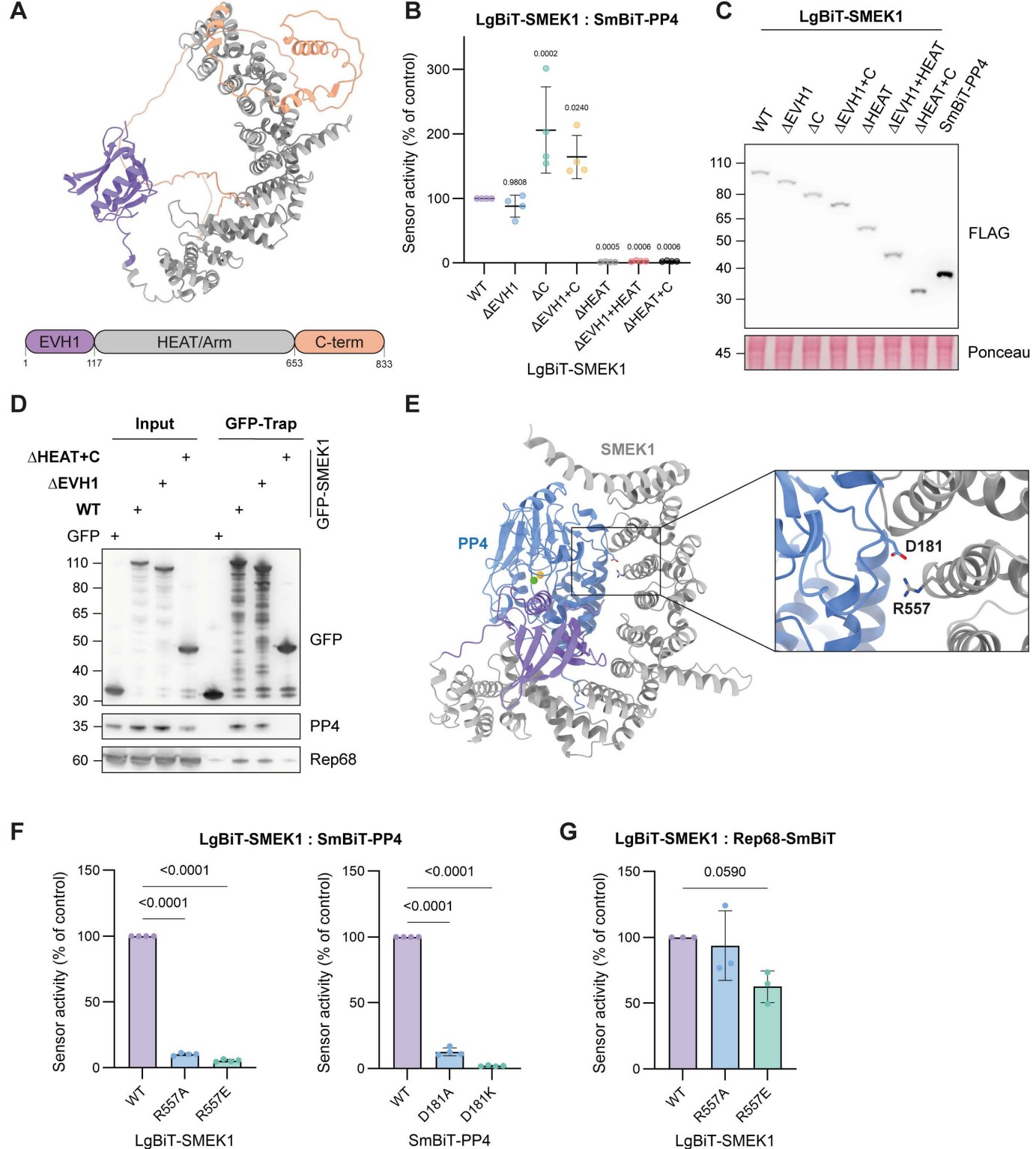

**Fig 2. Rep68 interacts with the HEAT/Arm repeat domain of SMEK1, independently of PP4. (A)** SMEK1 domain structure from the AlphaFold Protein Structure Database [57,58]. The EVH1, HEAT/Arm and C-terminal domain are indicated in purple, grey and orange respectively. **(B)** LgBiT-SMEK1 deletion mutants tested for their interaction with SmBiT-PP4. Lysate-based split-luciferase assay performed by mixing HEK293T lysates from cells

that separately expressed LgBiT-SMEK1 (WT or mutant) and SmBiT-PP4. Bioluminescence signal was measured after 20 minutes incubation at room temperature (end-point measurement). The data was plotted as a percentage of the LgBiT-SMEK1$^{WT}$:SmBiT-PP4 signal (mean ± SD; n = 4 independent experiments). Statistical significance was determined by an one-way ANOVA with Dunnett's multiple comparison test. **(C)** Transient expression levels of LgBiT-SMEK1 (WT or mutants) and SmBiT-PP4 assessed via immunoblotting of the HEK293T lysates used in Fig 2B. **(D)** GFP-trap results of ectopically expressed GFP-SMEK1 (WT and mutants) showing loss of binding between GFP-SMEK1$^{ΔHEAT+C}$ and PP4/Rep68. **(E)** AlphaFold 3 multimer prediction of the PP4:SMEK1 holoenzyme complex. The identified PP4$^{D181}$:SMEK1$^{R557}$ salt bridge is shown as sticks in the close-up panel. PP4 active site metals, $Zn^{2+}$ and $Fe^{2+}$, are represented as a green and orange sphere respectively. **(F)** Lysate-based split-luciferase end-point measurements of the LgBiT-SMEK1$^{R557→A/E}$:SmBiT-PP4$^{WT}$ (left panel) and LgBiT-SMEK1$^{WT}$:SmBiT-PP4$^{D181→A/K}$ (right panel) mutant interaction sensors. Bioluminescence signal was measured after 20 minutes of incubation at room temperature and plotted as a percentage of the LgBiT-SMEK1$^{WT}$:SmBiT-PP4$^{WT}$ signal (mean ± SD; n = 4 independent experiments). Statistical significance was determined by an one-way ANOVA with Dunnett's multiple comparison test. **(G)** Lysate-based split-luciferase end-point measurement of the LgBiT-SMEK1$^{R557→A/E}$:Rep68-SmBiT interaction sensors. Bioluminescence signal was measured after 20 minutes of incubation at room temperature. Data plotted as percentage of the LgBiT-SMEK1$^{WT}$:Rep68-SmBiT S/B signal. (mean ± SD; n = 3 independent repeats). Statistical significance was determined by an one-way ANOVA with Dunnett's multiple comparison test.

of the EVH1 domain did not affect this interaction (Fig 2B). As shown in Fig 2C, the loss of binding between SmBiT-PP4 and the different LgBiT-SMEK1$^{ΔHEAT}$ deletion mutants was not due to different expression levels of the mutants compared to LgBiT-SMEK1$^{WT}$. These results were confirmed by GFP-SMEK1 pull-downs (Fig 2D). Surprisingly, removal of the unstructured C-terminus enhanced the interaction of PP4 with the HEAT/Arm domain, indicating it somehow hinders the PP4 coordination by SMEK1. We also examined the purified SMEK1 deletion mutants for binding to purified Rep68-SmBit, using split-luciferase assays. Binding of Rep68 to SMEK1 was decreased after deletion of either the EVH1 or HEAT/Arm domain (S3B, S3C Fig). However, stronger loss of binding was observed in the SMEK1 deletion mutants where the HEAT/Arm domain was removed, indicating that Rep68 mainly interacts with the latter domain. Pull-down assays further confirmed that the SMEK1:Rep68 interaction is mainly mediated by the HEAT/Arm domain (Fig 2D).

Since the SMEK1:Rep68 and SMEK1:PP4 interaction both involve the HEAT/Arm domain of SMEK1, we subsequently investigated if PP4 association with SMEK1 is necessary for the SMEK1:Rep68 interaction. As there are no structural models for PP4:SMEK1, we used AlphaFold 3 to predict the structure of the PP4:SMEK1 complex [30]. This resulted in five models with high confidence scores, all converging on a single structural topology of the PP4:SMEK1 complex. Using this model, we could identify a salt bridge between SMEK1$^{R557}$ and PP4$^{D181}$ (Fig 2E). Mutation of these residues to alanine or its opposite charge largely abolished their interaction, as observed in both the lysate-based split-luciferase and FLAG-IP assays (Figs 2F and S4C). Mutation of SMEK1$^{R557}$ and PP4$^{D181}$ did not alter their expression levels (S4A Fig). Using thermal shift assays, we confirmed that loss of binding between LgBiT-SMEK1$^{R557A/E}$ and PP4 was not due to altered protein folding of SMEK1 (S4B Fig). The PP4-binding mutants of SMEK1 still interacted with Rep68 (Figs 2G and S4D), indicating that the interaction between Rep68 and SMEK1 is independent of PP4 within the complex. The interaction of FLAG-SMEK1 with SF3B1 was not affected upon R557 mutation, delivering additional proof that the FLAG-SMEK1$^{R557A/E}$ conformation was not altered (S4C Fig).

Collectively, these data show that the Rep proteins target the HEAT/Arm domain of SMEK1, independently of PP4.

## Removal of PP4 or its regulatory subunits enhances AAV replication and gene expression

We previously demonstrated that the transcriptional co-repressor KAP1 associates with the WT AAV genome to promote heterochromatin formation during AAV latency, thereby silencing AAV transcription and replication [5]. In contrast, a recombinant (r)AAV genome, that lacks the *rep* and *cap* ORFs but harbours the left and right inverted terminal repeat (ITR) was insensitive to KAP1-mediated silencing. Expression of the Rep proteins led to KAP1$^{S824}$ hyperphosphorylation, followed by AAV chromatin decondensation and, ultimately, AAV replication. We showed that this effect was independent of the AAV ITRs, Ad5 co-infection, and ATM activation (Figs 3A and S5A) [5]. Furthermore, we found that the Rep-mediated KAP1$^{S824}$ hyperphosphorylation was linked to their interaction with the PP1:NIPP1 phosphatase complex, which is involved in KAP1$^{S824}$ dephosphorylation. However, the precise mechanism by which the Rep proteins inhibit PP1:NIPP1-mediated dephosphorylation remains unidentified.

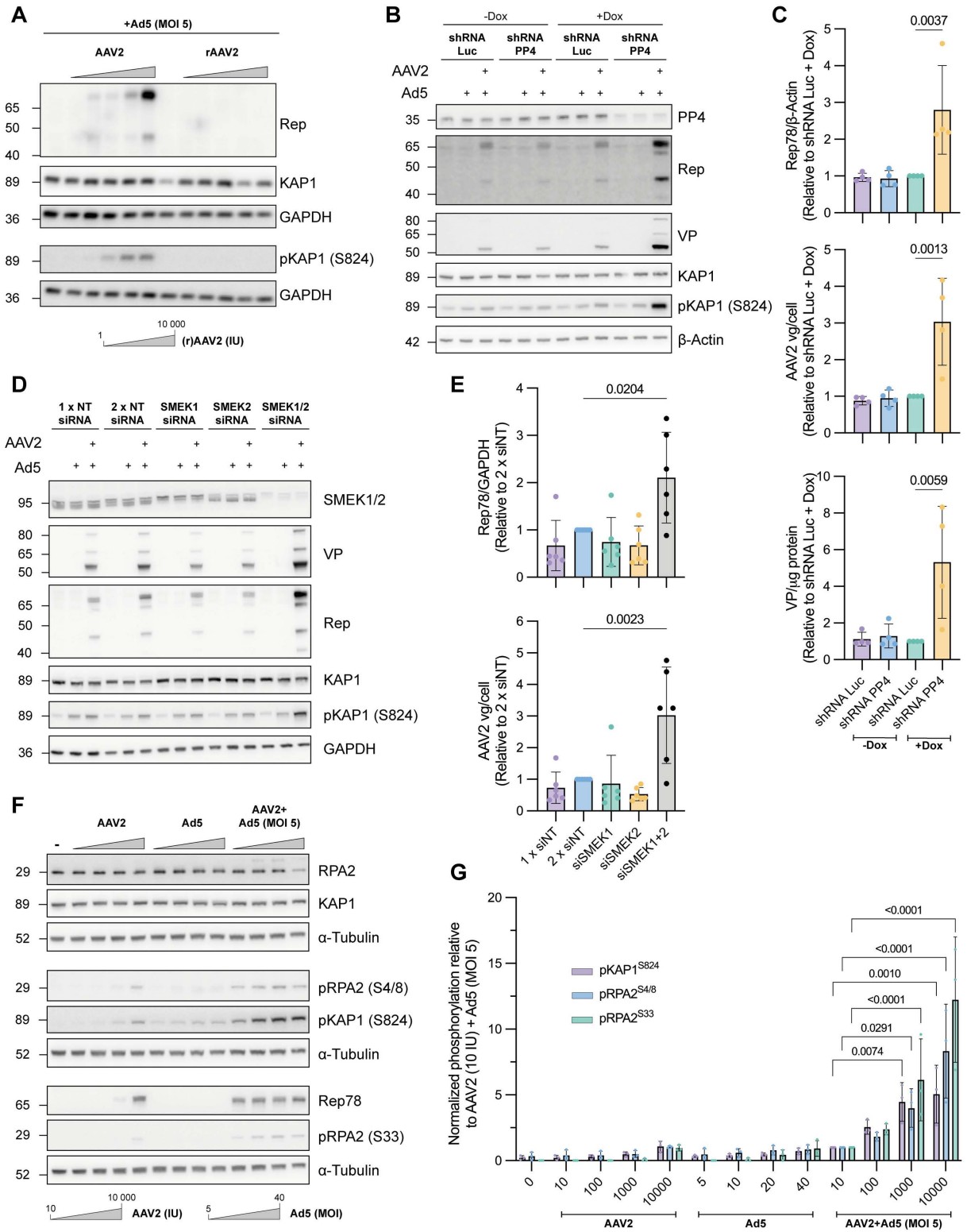

**Fig 3. PP4:SMEK1/2 holoenzyme activity represses AAV replication. (A)** Titration of increasing wild-type and recombinant AAV2 multiplicities of infection (MOIs) in HEK293T cells to assess the effect on KAP1$^{S824}$ phosphorylation. Cells were co-infected with Ad5 (MOI 5) and harvested 28h post-infection/transduction. Representative example of two independent repeats. **(B)** AAV replication assay in shRNA-mediated PP4C knockdown

HEK293T cell lines. Knockdown of PP4C was induced by treating the cells with 1 µg/mL doxycycline (Dox) for 96h after which they were infected with 10 IU AAV2 and a MOI 5 of Ad5. Cells were harvested 28h post-infection for further analysis. Shown are immunoblots of one representative example (n = 4 independent repeats). **(C)** From top to bottom: quantification of Rep expression levels via immunoblotting **(B)**; quantification of the AAV2 viral genomes per cell (vg/cell) via qPCR and quantification of the number of AAV2 viral particles per µg of total protein (VP/µg protein) via ELISA (mean ± SD; n = 4 independent experiments). Statistical significance was determined by an one-way ANOVA with Dunnett's multiple comparison test. **(D-E)** AAV replication assay after siRNA-mediated depletion of SMEK1 and SMEK2. siRNA against SMEK1 and SMEK2 were transiently transfected in HEK293T. SMEK1/2 depleted cells were infected 72h post-transfection with 10 IU AAV2 and a MOI 5 of Ad5. Cells were harvested 28h post-infection and processed as in **B-C** (mean ± SD; n = 6 independent experiments). Statistical significance was determined by an one-way ANOVA with Tukey's multiple comparison test. **(F)** Effect of lytic AAV2 infection on the phosphorylation status of KAP1$^{S824}$, RPA2$^{S4/8}$ and RPA2$^{S33}$. HEK293T cells were infected with either increasing IUs of AAV2, Ad5 or AAV2 with a fixed Ad5 MOI of 5. **(G)** Quantification of pKAP1$^{S824}$, pRPA2$^{S4/8}$ and pRPA2$^{S33}$ in F (mean ± SD; n = 3 independent experiments). Statistical significance was determined by a two-way ANOVA with Dunnett's multiple comparison test.

Because the AAV Rep proteins interact directly with SMEK1, we speculated that they might not only interfere with PP1 phosphatase activity, but also with PP4 activity to prevent dephosphorylation of substrates such as, e.g., KAP1 that otherwise hinder viral replication. To examine whether the PP4 phosphatase affects AAV replication, we generated a doxycycline-inducible shRNA-mediated PP4 knockdown (KD) HEK293T cell line. This cell line contains a stable genomic integration of a sequence encoding a shRNA that targets the 3'-UTR of *PPP4C* (shRNA PP4), under the control of a doxycycline-inducible promoter. Similarly, a control cell line was made with the stable genomic integration of a scrambled shRNA targeting the coding sequence of the firefly luciferase (shRNA Luc). We found that doxycycline-induced knockdown of PP4 was associated with enhanced viral protein expression levels, increased AAV replication and viral particle (VP) production (Fig 3B, 3C). The remaining PP4 levels in the replication assays were 32.3% (S5B Fig). Furthermore, knockdown of PP4 resulted in a marked increase in KAP1 phosphorylation, while KAP1 expression levels remained unchanged (Fig 3B).

Since Rep interacts directly with SMEK1, we also carried out siRNA-mediated knockdown experiments to assess the effect of SMEK1 depletion on AAV replication. Surprisingly, KD of SMEK1 alone had no discernible effect on viral replication (Fig 3D, 3E). Given the extended sequence and structural homology between SMEK1 and SMEK2 (S6A, S6B Fig), we next investigated whether simultaneous knockdown of both subunits would mimic the effect observed with PP4 knockdown. Indeed, the combined knockdown of SMEK1 and SMEK2, resulting in residual protein levels of 19.7%, significantly enhanced viral replication and gene expression, indicating functional redundancy between these regulatory subunits (Figs. 3D, 3E and S5C). This functional redundancy is consistent with their high structural similarity and is further supported by CRISPR-based dependency data from the DepMap project, which show that SMEK1 and SMEK2 are functionally co-dependent in a wide range of cell lines (https://depmap.org/portal/) (S6C Fig). In view of the structural and functional similarity between SMEK1 and SMEK2, we next examined whether Rep proteins also interact with SMEK2. GFP-trap assays using GFP-tagged SMEK1 and SMEK2 revealed robust co-precipitation of Rep68 with both SMEK proteins, indicating that Rep68 can associate with either regulatory subunit (S6D Fig).

Finally, we observed that infection of HEK293T cells with increasing AAV infection units (IU) in the presence of a fixed multiplicity of infection (MOI) of Ad5, led to a marked increase in phosphorylation of RPA2 at residues Ser4, Ser8 and Ser33 which are well-established targets of PP4 (Fig 3F, 3G) [22]. In contrast, Ad5-only infection did not induce RPA2 phosphorylation, indicating that this hyperphosphorylation is specifically mediated by the AAV Rep proteins. These findings are in line with our previous observation of elevated KAP1$^{S824}$ phosphorylation during lytic AAV infection (Fig 3F, 3G) [5].

Together, these findings demonstrate that the PP4:SMEK1 and PP4:SMEK2 holoenzymes function as negative regulators of AAV replication and gene expression. This repression is most likely alleviated through Rep-dependent inhibition of PP4 phosphatase activity, through a sustained hyperphosphorylation of PP4 substrates such as KAP1 and RPA2.

## The PP4 substrate KAP1 interacts with SMEK1 and SMEK2 via its MAPP short linear motif

To better understand how KAP1 is recruited to PP4 via its regulatory subunits, we investigated the specific interaction sites between KAP1 and SMEK1/2. PP4:SMEK1/2 substrates usually harbour a conserved FxxP/MxPP SLiM motif, which binds with low micromolar affinity to a hydrophobic groove in the EVH1 domain of SMEK1/2 [20]. KAP1 contains a single such motif, 423MAPP426 (Fig 4A). SLiMs are typically characterized by two or three key residues located within a sequence of up to ten amino acids, often found in intrinsically disordered regions of the proteins involved in the interaction [31]. The 423MAPP426 SLiM motif in KAP1 is located N-terminal to the Ser473 and Ser824 residues, which are phosphorylated by Chk1/Chk2 and ATM, respectively, during the DNA damage response, leading to chromatin relaxation. Following DNA repair, Ser473 and Ser824 residues are dephosphorylated by PP4 [19,32]. The AlphaFold-predicted structure of KAP1 suggests that its 423MAPP426 motif resides within an intrinsically disordered region (Fig 4A). Alanine substitution of this motif (423MAPP426 → 423AAAA426) (SmBiT-KAP1MAPP) strongly reduced its interaction with LgBiT-SMEK1 in lysate-based split-luciferase assays (Fig 4B) without affecting its expression levels compared to SmBiT-KAP1WT (S7A Fig). Additionally, deletion of the conserved EVH1 domain from SMEK1 decreased its interaction with KAP1 (S7A, S7B Fig), indicating that the MAPP motif of KAP1 interacts with the EVH1 motif of SMEK1. GFP-Trap assays of GFP-SMEK1 to assess co-precipitation of FLAG-tagged KAP1WT/MAPP confirmed the reduced affinity between GFP-SMEK1WT and FLAG-KAP1MAPP, and between GFP-SMEK1ΔEVH1 and FLAG-KAP1WT (S7C Fig).

The interaction of the KAP1 MAPP motif with SMEK1's EVH1 domain was further validated using AlphaFold 3 multimer predictions. Modeling of the SMEK1 EVH1 domain with a KAP1413-437-derived peptide containing the MAPP motif yielded five high-confidence models that converged on a single topology. In all models, the peptide was positioned within the hydrophobic groove of the EVH1 domain, resembling the binding mode of endogenous PP4 substrates that engage SMEK1/2 via an FxxP SLiM (Fig 4C) [20]. Structural alignment of the SMEK1EVH1:KAP1MAPP AlphaFold model with the published crystal structure of SMEK1EVH1 in complex with a model FxxP peptide revealed that both peptides aligned with a RMSD of 0.539 (S7D Fig) [20].

We next utilized the EVH1-binding peptide (SLPFT<u>FKVP</u>APPPSLPPS) described by Ueki et al. for competition studies [20]. We fused four copies of this peptide to YFP (YFP-4xFKVP) (Fig 4D), as it was previously shown that this multivalency significantly improves the binding affinity of SLiMs, enhancing its potency as a competitive inhibitor of SMEK1/2 substrate interactions [33]. The YFP-4xFKVP competitor peptide effectively disrupted the SMEK1:KAP1 interaction in lysate-based split-luciferase assays (Figs 4E, 4F and S7E). A similar effect was observed for the LgBiT-SMEK2:SmBiT-KAP1 interaction sensor (S7F Fig). In contrast, a control peptide (YFP-4xAKVA) barely affected the SMEK1/2:KAP1 interaction.

In summary, we were able to disrupt the SMEK1:KAP1 interaction by mutating the 423MAPP426 motif in KAP1, deleting the EVH1 domain of SMEK1 or through competitive displacement by adding the YFP-4xFKVP competitor, demonstrating that KAP1 binds the EVH1 domain of SMEK1 at the canonical 'FxxP'-substrate interaction site via its 423MAPP426 SLiM.

## Rep68 prevents the recruitment of PP4 substrates via SMEK1/2

The observation that lytic AAV infection, and the associated expression of Rep proteins, results in increased phosphorylation of PP4 substrates (Fig 3F), led us to speculate that Rep proteins inhibit PP4 holoenzymes. One possible mechanism is through interference with substrate recruitment. We performed various lysate-based split-luciferase assays to test whether Rep68 interferes with substrate recruitment of the PP4:SMEK1 holoenzyme. First, we observed that the purified KAP1 competes with Rep68-SmBiT for binding to LgBiT-SMEK1 (Fig 5A). Conversely, addition of purified Rep68 also competed with SmBiT-KAP1 binding to LgBiT-SMEK1 (Fig 5B). This indicates that both KAP1 and Rep68 share overlapping interaction sites on SMEK1.

To further validate the competition between KAP1 and Rep68 for binding to SMEK1, we used the YFP-4xFKVP peptide as a competitor in our lysate-based split-luciferase assays (Fig 5C). Addition of the YFP-4xFKVP competitor peptide to the

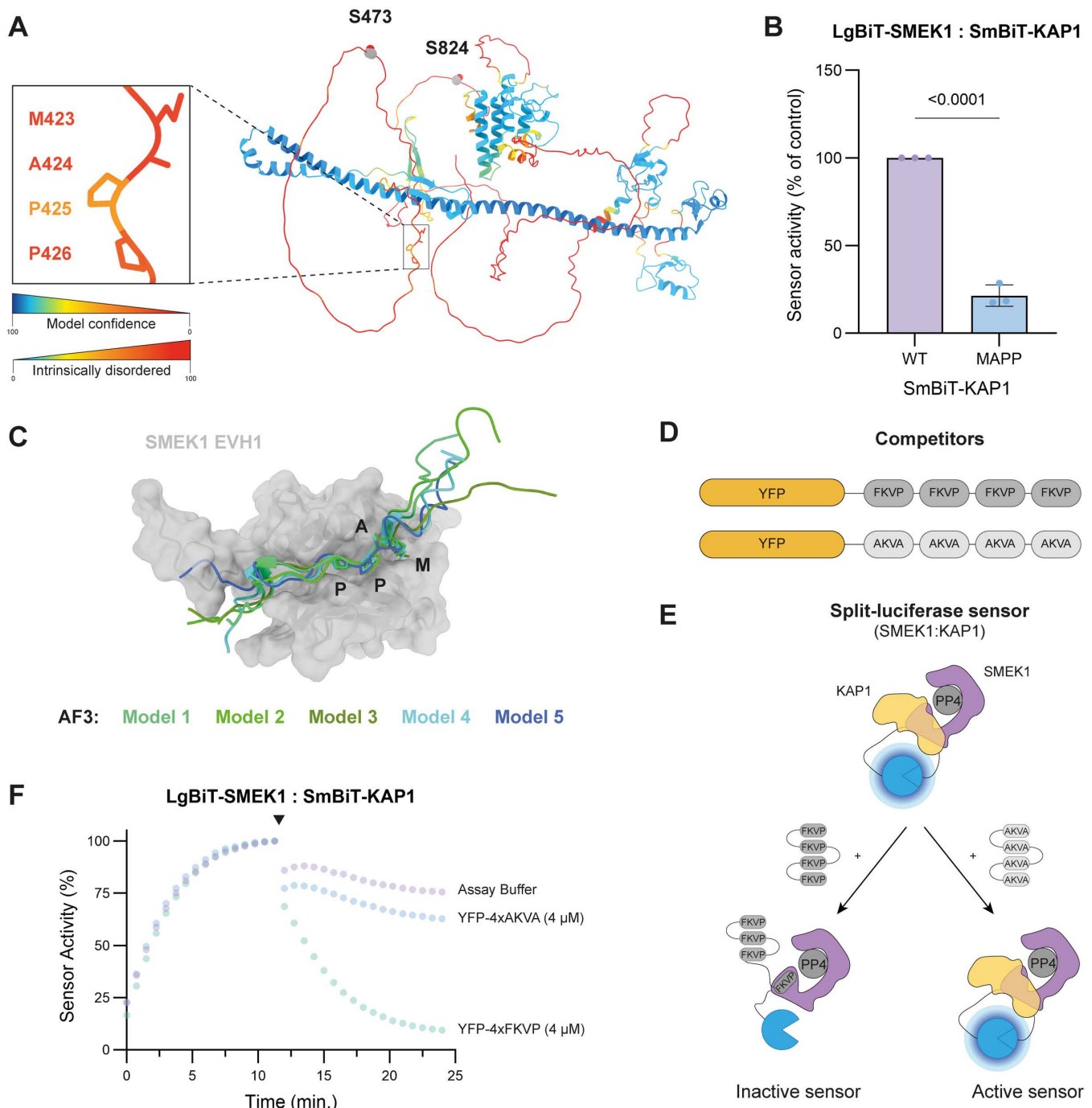

**Fig 4. KAP1 interacts via its MAPP SLiM with the EVH1 domain of SMEK1 and SMEK2. (A)** KAP1 (TRIM28) tertiary structure modelled via the AlphaFold Protein Structure Database. The structure is coloured according to their local model confidence. The lower the model confidence, the higher the intrinsically disordered character. MxPP SLiM ([423]MAPP[426]) highlighted as sticks in the close-up view (left panel). **(B)** Lysate-based split-luciferase assay (end-point measurement) of the LgBiT-SMEK1:SmBiT-KAP1[MAPP] interaction sensor. Bioluminescence signal was measured after 20 minutes of incubation at room temperature (end-point measurement). The data was plotted as a percentage of the LgBiT-SMEK1[WT]:SmBiT-KAP1[WT] signal (mean±SD; n=3 independent experiments). Statistical significance was determined by a two-tailed unpaired t-test. **(C)** AlphaFold 3 multimer prediction of the SMEK1[EVH1] domain in complex with the KAP1[413-437] peptide containing the [423]MAPP[426] SLiM. The five obtained models were structurally aligned. **(D)** Design of the YFP-4xFKVP competitor peptides. YFP-4xAKVA serves as the non-binding control peptide. **(E)** Design of the kinetic-trace experiments using the LgBiT-SMEK1:SmBiT-KAP1 interaction sensor and YFP-4xFKVP as competitor. For simplicity, the YFP-tagged competitor and control peptide

are represented without YFP. **(F)** Kinetic-trace experiment showing the time-dependent association of the LgBiT-SMEK1:SmBiT-KAP1 split-luciferase sensor. The black arrow indicates the addition of the purified YFP-4xFKVP/AKVA competitor (concentration indicated in the graph). The represented data is plotted as a percentage of the S/B ratio right before the addition of the competitor. The data shown is a representative example of three independent repeats.

LgBiT-SMEK1:Rep68-SmBiT interaction sensors resulted in dissociation of the protein complex, while the control peptide had no significant effect on the interaction (Fig 5C). This confirms the data obtained in S3B Fig, showing that Rep68 interacts with the EVH1 domain of SMEK1, and thereby probably interferes with substrate recruitment. However, previous pull-down and split-luciferase experiments (Figs 2D and S3B) showed that the HEAT/Arm repeat domain of SMEK1 is the main determinant for the SMEK1:Rep68 interaction. The YFP-4xFKVP peptide produced similar effects when added to the LgBiT-SMEK2:Rep68-SmBiT interaction sensor (S8A Fig).

To confirm the specificity of the effect of the FKVP peptide on EVH1-mediated interactions, we performed a control experiment using the LgBiT-tagged B56$_\delta$ regulatory subunit of PP2A and SmBiT-RepoMan, a known PP2A substrate. RepoMan recruitment to PP2A via B56$_\delta$ depends on the LSPI SLiM and thus should not be affected by the YFP-4xFKVP peptide. The PP4-specific SLiM peptide showed no effect on the B56$_\delta$:RepoMan interaction, whereas the YFP-4xLSPI peptide almost completely dissociated the interaction sensor (S8B Fig), confirming the selectivity of the YFP-4xFKVP competitor peptide for EVH1 binding substrates [20].

In conclusion, these data show that Rep68 and KAP1 may share overlapping binding sites on SMEK1, or that steric hinderance upon SMEK1 binding of the viral protein prevents the binding of the other PP4 substrates. Furthermore, we demonstrate that the interaction of both proteins with SMEK1 are at least in part mediated by the FxxP docking site on the EVH1 domain of SMEK1/2.

## The NIPP1 regulatory subunit of PP1 interacts with the SMEK1 regulatory subunit of PP4

We previously reported that AAV Rep proteins also target the PP1:NIPP1 holoenzyme complex to promote viral replication and transcription [5]. This prompted us to explore whether PP1:NIPP1 and PP4:SMEK1 are part of the same macromolecular complex. For this reason, we performed FLAG-IP experiments from cells that were transiently transfected with FLAG-tagged full-length NIPP1$^{WT}$, NIPP1$^{Δ1-22}$ lacking the first 22 amino acids known as the FHA-inhibitory domain (FID) [26] or NIPP1$^{Δ1-142}$ lacking the FID and substrate-binding FHA domain (Fig 6A). As expected, binding of Rep68 and the substrates KAP1 and SF3B1 to NIPP1, depend on the N-terminal FHA domain of NIPP1 [5,23]. Surprisingly, we also observed an association between endogenous SMEK1 and FLAG-NIPP1$^{WT}$. This association was not affected by deletion of the FID domain (FLAG-NIPP1$^{Δ1-22}$) but was abolished by the additional deletion of the FHA domain (NIPP1$^{Δ1-142}$) (Fig 6A). This was confirmed by a GFP-trap of GFP-SMEK1 from cells ectopically expressing either FLAG-tagged wild-type or NIPP1$^{Δ1-142}$ (Fig 6B). Additional GFP-trap experiments with truncated mutants of GFP-SMEK1 indicated that the HEAT/Arm domain of SMEK1 is important for the association with endogenous NIPP1 (S9A Fig).

NIPP1 recruits substrates for associated PP1 through its FHA domain, which recognizes a phosphorylated threonine-proline (pTP) motif on the substrate, where the threonine is phosphorylated by cyclin-dependent kinases (CDKs). Known PP1:NIPP1 substrates that interact via this mechanism include SF3B1, CDC5L, MELK, EZH2 [23,34,35]. We identified a TP-dipeptide motif ($^{277}$TP$^{278}$) in SMEK1, located in a disordered connecting loop in the HEAT/Arm domain (Fig 6C). A GFP-trap was performed using GFP-SMEK1 in which the TP-dipeptide motif was mutated to AA (phosphomutant) or DP (phosphomimetic). The TP→AA mutant of SMEK1 resulted in a slightly decreased association with NIPP1, while no strengthened binding of the TP→DP mutant was observed (Fig 6D). This might be due to the unknown extent of phosphorylation in the WT construct and uncertainty whether mutation to aspartic acid effectively mimics phosphorylation on this site in the TP→DP mutant. Possibly, the bulkiness of a phosphate group is required to support proper interaction with the FHA domain of NIPP1.

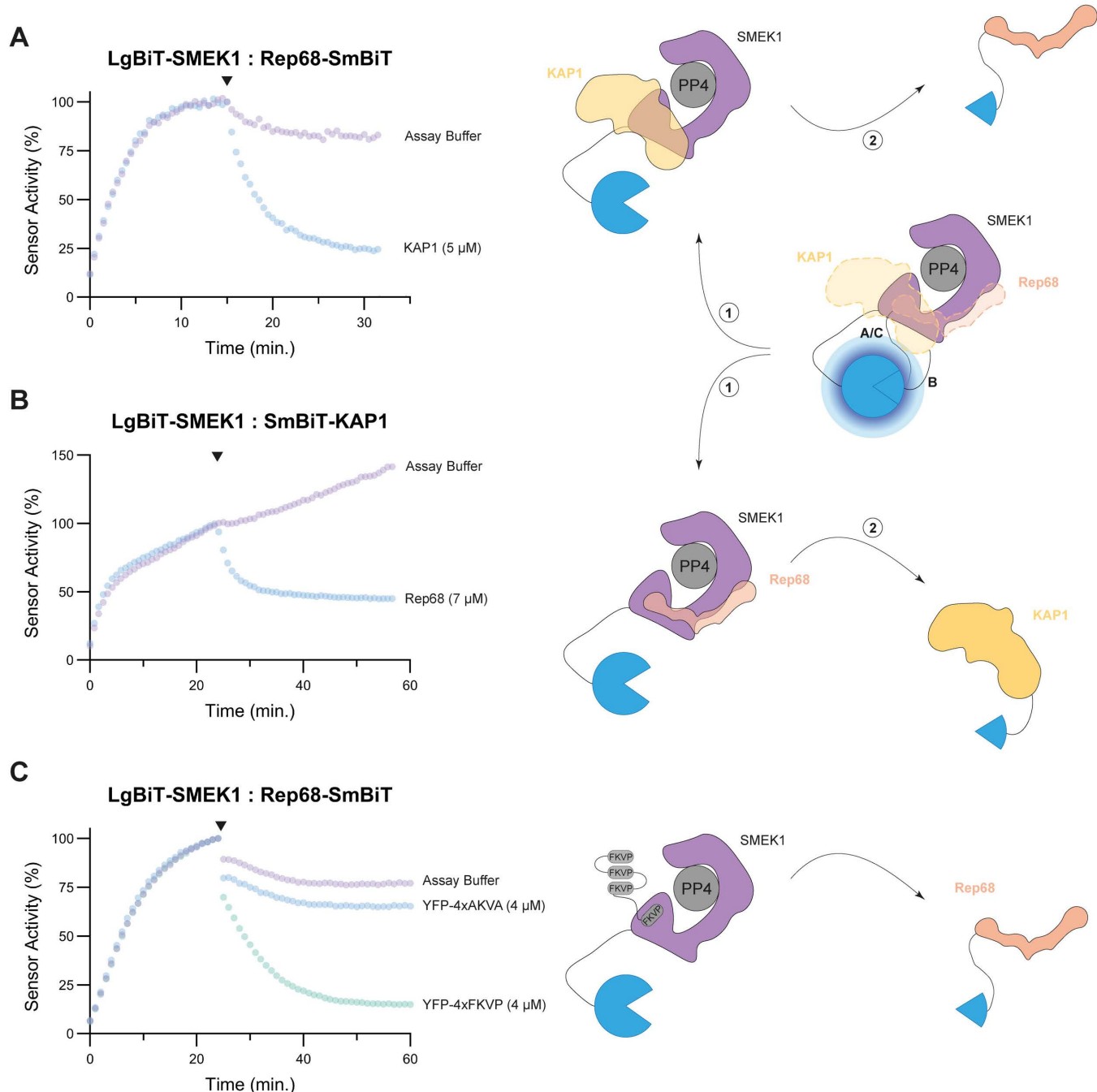

**Fig 5. AAV Rep68 interferes with the recruitment of PP4:SMEK1 substrates. (A)** Kinetic-trace experiment showing the time-dependent association of the LgBiT-SMEK1:Rep68-SmBiT split-luciferase sensor. The black arrow indicates the addition of the purified, untagged KAP1 competitor (concentration indicated in the graph). The represented data is plotted as a percentage of the S/B ratio right before the addition of the competitor. The data shown is a representative example of three independent repeats. **(B)** Same experiment as in **A**, but with the LgBiT-SMEK1:SmBiT-KAP1 split-luciferase sensor and purified, untagged Rep68 as competitor. **(C)** Analogous experiment as in **A**, but with YFP-4xFKVP/AKVA as competitor.

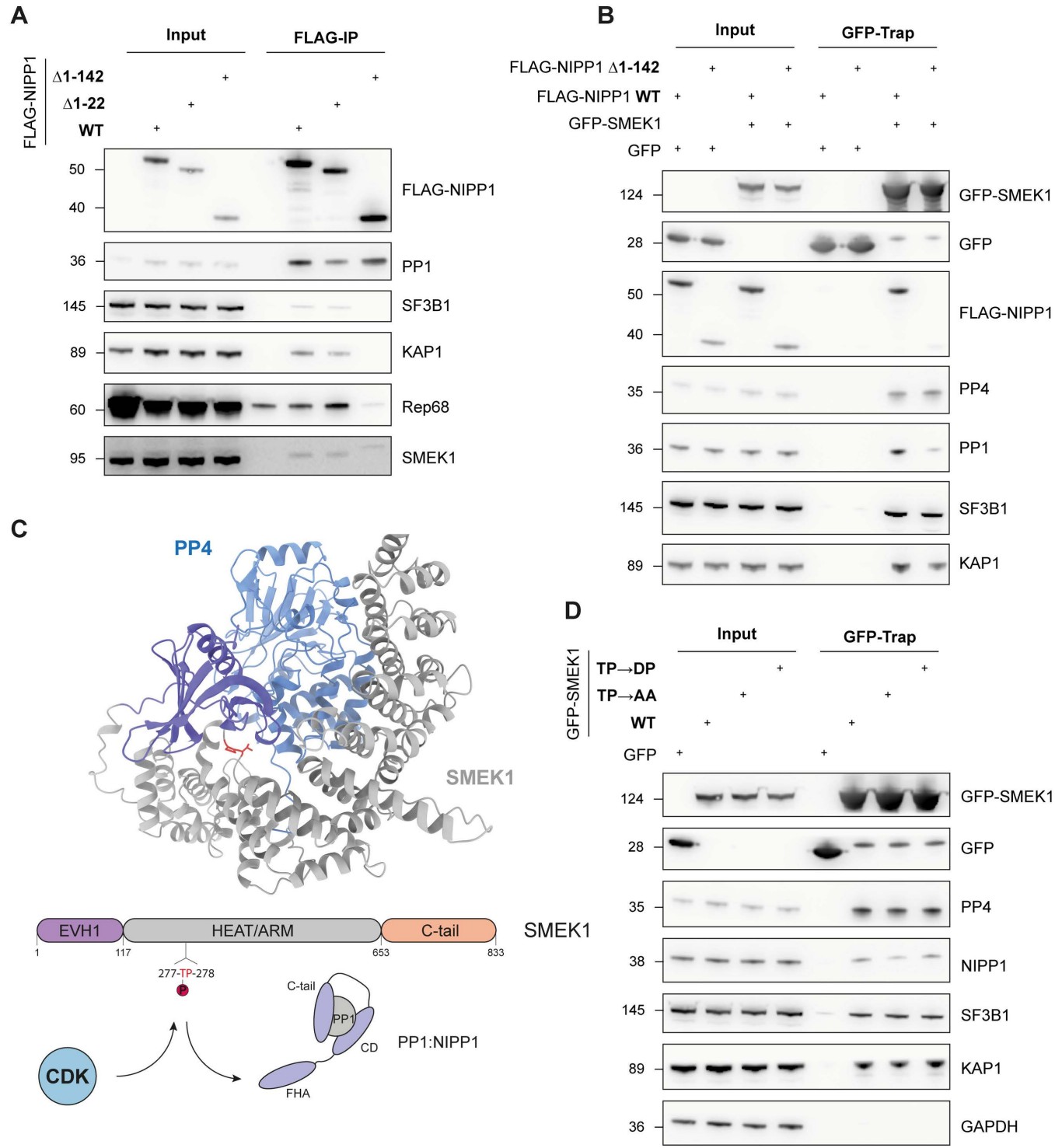

**Fig 6. Rep interacts with the putative PP1:NIPP1-PP4:SMEK1 multifunctional phosphatase complex. (A)** FLAG-IP of FLAG-NIPP1 (WT, Δ1-22 and Δ1-142) transiently expressed in HEK293T. **(B)** GFP-trap of GFP-SMEK1 to investigate the co-precipitation of FLAG-NIPP1 (WT and Δ1-142) transiently expressed in HEK293T cells. **(C)** AlphaFlold 3 model of the PP4:SMEK1 complex, with the SMEK1^TP dipeptide motif highlighted in red as sticks. CDK kinases might phosphorylate SMEK1$^{T277}$, thereby increasing the affinity of SMEK1 for NIPP1$^{FHA}$ binding. **(D)** GFP-trap of GFP-SMEK1 (TP→AA or DP) to investigate the effect on co-precipitation of endogenous NIPP1.

Together with our previous observations [5], the above data indicate that PP1:NIPP1 and PP4:SMEK1 are components of the same macromolecular complex, which are targeted by AAV Rep proteins. The interaction between SMEK1 and the NIPP1 FHA domain is likely phosphorylation-independent.

## Discussion

Here, we provide for the first time evidence that a viral protein directly engages the PP4:SMEK1/2 phosphatase holoenzyme to favour viral propagation. Our findings reveal that this complex acts as a negative regulator of AAV replication and gene expression, and that its repressive activity is antagonized by the Rep proteins through a novel viral substrate-recruitment interference mechanism that serves to retain PP4 substrates in a hyperphosphorylated state (Fig 7). One of these substrates is KAP1, whose activity our group previously linked to AAV chromatinization [5].

SMEK1 (PP4R3A) was identified as a candidate interactor of AAV Rep proteins [5], prompting us to examine whether Rep proteins also target PP4 holoenzymes. To our knowledge, no prior study has reported a viral protein interacting with the trimeric PP4:PP4R2:SMEK1/2 complex. While the MCPyV small-T antigen is known to associate with PP4:PP4R1 and to modulate NF-κB signalling [36,37], our study characterizes a novel interaction between Rep proteins and PP4:SMEK1/2. We have found that the AAA$^+$-ATPase/helicase domain, conserved across all four Rep isoforms, mainly interacts with the HEAT/Arm domain of SMEK1, a region previously shown to coordinate the PP4 catalytic subunit (S1C and S3B Figs) [28]. AlphaFold3 modelling of the PP4:SMEK1 complex further supports this interaction, identifying a single salt bridge (PP4$^{D181}$:SMEK1$^{R557}$) as essential for complex formation (Fig 2F).

Substrates of the PP4:SMEK1/2 holoenzyme typically harbour a FxxP or MxPP SLiM within a disordered region that binds with low micromolar affinity in a conserved hydrophobic groove of the SMEK1 EVH1 domain [20]. Based on this

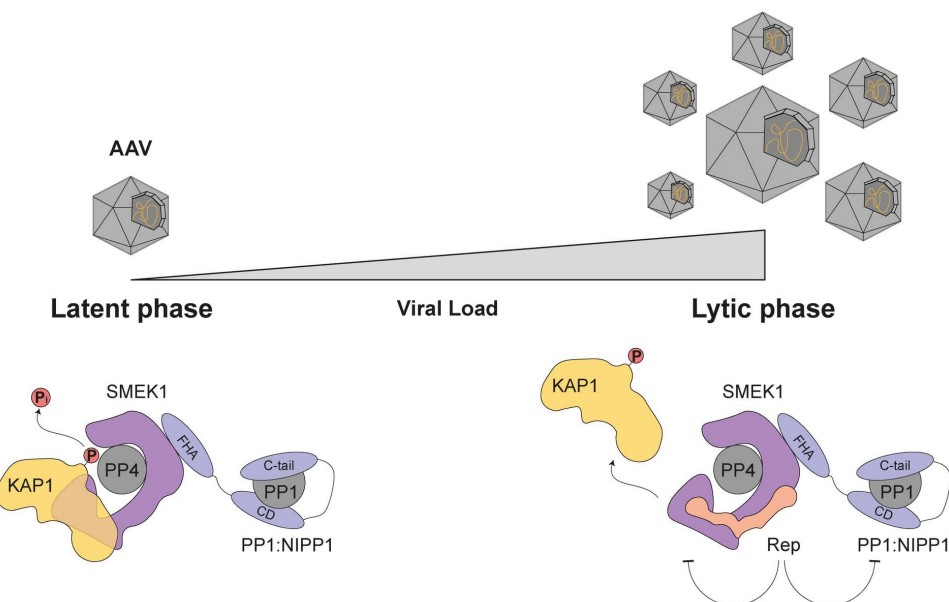

**Fig 7. Model of the Rep-mediated substrate interference mechanism.** During a latent AAV infection, in the absence of Rep expression, PP4:SMEK1 holoenzymes are active. They actively dephosphorylate substrates that are involved in the AAV life cycle such as KAP1 and keep the AAV genome silenced. The switch from the latent to the lytic phase of the viral life cycle, upon superinfection with a helper virus, causes the expression of the viral Rep proteins, hereby boosting viral replication. The Rep proteins, together with viral replication, elicit a DNA damage response (DDR), hyperphosphorylating DDR-related proteins such as KAP1 and RPA2. The Rep proteins interact with the PP4:SMEK1 and PP1:NIPP1 holoenzymes, resulting in inhibition of PP4/PP1 phosphatase activity through interference with the recruitment of PP4:SMEK1 holoenzyme substrates. This inhibition maintains substrate hyperphosphorylation, thereby preventing repression of AAV gene expression and replication.

knowledge, we identified a similar motif ([423]MAPP[426]) in KAP1 (Fig 4A). Mutation of this motif or deletion of the EVH1 domain significantly weakened the interaction of KAP1 with SMEK1, although residual binding suggests additional contact sites. However, GFP-trap experiments (S7C Fig) indicated that either deletion of the SMEK1 EVH1 domain or mutation of the KAP1 MAPP motif completely abolished the interaction with the WT interaction partner. This exemplifies the higher sensitivity of split-luciferase assays for studying lower affinity protein-protein interactions. AlphaFold3 docking of the KAP1 MAPP motif on EVH1 yields five high-confidence models that all converge on a similar binding topology as observed in the EVH1:FxxP crystal structure (Figs 4C and S7D) [20].

Intriguingly, the AAV Rep proteins compete with endogenous, FxxP/MxPP-containing PP4:SMEK1/2 substrates such as KAP1 for SMEK1 binding, thereby blocking substrate recruitment. The hypothesis of substrate-recruitment interference by the Rep proteins was further supported by affinity measurements (Fig 1G). MST assays revealed that SMEK1 binds Rep68 with a slightly higher affinity ($K_D = 453$ nM) than KAP1 ($K_D = 776$ nM). Additional research is needed to identify the residues in Rep that are involved in the interaction with SMEK1 and to examine whether Rep proteins also function as a substrate for PP4-mediated dephosphorylation [38,39]. In this respect, it is worth noting that Rep proteins are post-translationally modified by phosphorylation [38]. Dephosphorylation of Rep proteins was suggested to be mediated by PP2A [39], but the underlying experiments are not conclusive as they were largely based on the use of okadaic acid (100 nM), which is not specific for PP2A but also inhibits PP4 [40].

Certain viral proteins target the PP2A:B56 holoenzyme using a conserved LxxIxE SLiM [41,42], similar to how AAV Rep proteins disrupt PP4:SMEK1/2 interactions. For example, HTLV-1 integrase (IN) binds B56γ via this SLiM [43,44], potentially recruiting PP2A:B56γ to integration sites to dephosphorylate transcription factors or chromatin modifiers. EBOV nucleoprotein (NP) uses a similar strategy, guiding VP30 to PP2A:B56 for dephosphorylation via a B56 SLiM, causing the shift from replication to transcription [3]. While Ebola virus and HTLV-1 exploit PP2A activity, AAV instead inhibits PP4 to promote its replication and gene expression. This Rep-mediated substrate-recruitment interference represents a novel viral strategy targeting PPP-type phosphatases. One hypothesis could be that Rep proteins act as unfair competitors that have extremely slow dephosphorylation rates, hereby blocking the recruitment of endogenous substrates. This unfair competition mechanism has previously been characterized for both PP1 and PP2A [45,46].

Beside KAP1[S824] hyperphosphorylation we also observe RPA2[S4/8/33] hyper-phosphorylation, another PP4:SMEK substrate previously linked to AAV (Fig 3F, 3G) [47]. In addition to understanding the role of RPA2 in the AAV life cycle, further investigation is needed to unravel the consequence of Rep expression for the global human phosphoproteome and for specific PP4:PP4R2:SMEK1/2 substrates (e.g. γH2AX, DBC1 and 53 BP1) [21,48,49]. However, complete PP4 inhibition is unlikely, as this is known to be lethal ([https://depmap.org/portal/](https://depmap.org/portal/)). It is possible that some PP4:SMEK1 substrates do not solely rely on an FxxP/MxPP SLiM, similar to PP2A:B56 substrates that can bypass LxxIxE-mediated recruitment [50]. Therefore, partial PP4 activity may persist despite Rep binding. This highlights that AAV must rewire, rather than abolish PP4 function to preserve host cell viability and support viral replication.

Remarkably, our data also revealed that SMEK1 interacts with NIPP1 and one of its substrates SF3B1 (Figs 1B and 6A, 6B), indicating that SMEK1 may function as a scaffold for the assembly of a novel multifunctional PP1:PP4 complex. This multifunctional PP1:PP4 complex may be part of the larger spliceosomal assembly, as high-throughput interaction studies reveal extensive associations between SMEK1 and multiple spliceosomal components, including SF3B1 [51]. Several multifunctional protein complexes with phosphatases have been described before. Protein kinase A-anchoring proteins (AKAPs) bind with PKA and other kinases, thereby targeting these kinases to their substrates [52]. Phosphatases like PP2A and PP2B can be part of such complexes to dynamically antagonize kinase signalling [53–55]. Furthermore, one study has reported on a PP1:PP2A phosphatase complex in yeast, being essential for the re-activation of both PP1 and PP2A in late mitosis [56]. Here, we provide evidence for the existence of a PP1:PP4 complex that involves the HEAT/Arm domain of SMEK1 and the FHA domain of NIPP1 (Fig 7). Substrates of PP1:NIPP1 often interact via a phosphorylated TP dipeptide motif [23,34,35]. However, we found no evidence that SMEK1/2 binds to NIPP1 as substrate, indicating

the SMEK1 coordinates the function of both associated PP4 and PP1. Given that both PP1:NIPP1 and PP4:SMEK1 are implicated in KAP1[S824] dephosphorylation, this raises the intriguing possibility that Rep proteins hamper the dephosphorylation of KAP1, and possibly other substrates, by both PP1 and PP4. (Fig 7).

In conclusion, our findings reveal a novel mechanism of PPP-type phosphatase inhibition via viral-mediated substrate-recruitment interference. Furthermore, we demonstrate that the transcriptional co-repressor KAP1 primarily interacts with the EVH1 domain of SMEK1 via its MAPP SLiM. Notably, hyperphosphorylation of several repressive proteins, including KAP1 and RPA2, emerges as a recurring feature during lytic AAV infection and Rep expression. Through Rep-mediated inhibition of PP4 described here, we further illustrate the sophisticated strategy employed by Rep proteins to sustain the hyperphosphorylation of key PP4 substrates. This prevents the activation of host repressive mechanisms that would otherwise hinder viral replication, while simultaneously preserving cellular pathways that are advantageous to the viral life cycle. These insights extend our understanding of how AAV hijacks host regulatory networks and may inform strategies to enhance the efficacy and manufacturability of rAAV vectors. Further studies will be required to determine whether phosphatases and their substrates also play a role in regulating therapeutic gene expression or contribute to the complex host-cell interactions underlying rAAV production. Such knowledge could uncover new molecular targets for engineering more potent vectors and optimizing clinical gene therapy manufacturing platforms. In parallel, the growing interest in targeting phosphatases for therapeutic purposes highlights the broader relevance of this work, which may offer new insights into how phosphatase activity can be modulated to treat diseases such as cancer.

## Materials and methods

### Protein expression and purification

His-Rep68, His-KAP1, His-LgBiT-SMEK1 (WT and deletion mutants), His-Rep68-SmBiT, His-SmBiT-KAP1, His-YFP-4xFKVP, His-YFP-4xAKVA and His-YFP-4xLSPI were expressed from the pET16b vector in BL21 Gold *E. coli* cells. Cells were grown in Luria Bertani broth, supplemented with 100 µg/mL ampicillin, at 37°C until the optical density at 600 nm reached 0.6-0.7. Next, recombinant protein expression was induced by adding 1 mM IPTG to the cell suspension. Protein expression was induced overnight at 18°C. Cells were pelleted by centrifugation (6000 x g, 10 min., 4°C) and resuspended in lysis buffer containing 50 mM Tris at pH 7.9, 0.5 NaCl, 10% glycerol, 0.5% Triton X-100, 1 mM phenylmethylsulfonyl fluoride (PMSF), 1 mM benzamidine and 0.5 µg/mL leupeptin (LPP). Cell suspensions were subjected to one freeze-thaw cycle, followed by incubation with 10 U benzonase and 0.5 mg lysozymes per mL cell lysate for 20 minutes at room temperature. After the benzonase and lysozyme treatment, the lysates were sonicated for 15 minutes with a Diagenode Bioruptor sonicator (15 min., 15 sec. ON, 15 sec. OFF). The debris was spun down (15 000 x g, 20 min., 4°C) and cleared lysates were incubated with Ni$^{2+}$-Sepharose beads for 1 h at 4°C. The beads were washed two times with lysis buffer and afterwards with buffer containing 50 mM Tris at pH 7.9, 0.5 M NaCl and imidazole concentrations of either 20 or 60 mM. Proteins were eluted from the beads by applying the same Tris-based buffer to the beads containing 0.4 M imidazole. Eluted proteins were dialyzed to a buffer containing 50 mM Tris at pH 7.5 and 150 mM NaCl. Purity of the purified proteins was confirmed by SDS-PAGE and Coomassie staining. Concentration of the purified proteins was determined via NanoDrop spectrophotometer measurements of the absorbance at 280 nm.

### Cell culture and treatments

HEK293T cells were cultured in Dulbecco's modified Eagle's medium (Gibco) with 4.5 mg glucose/mL, supplemented with 10% fetal calf serum (Sigma-Aldrich), 100 units penicillin (P)/mL and 100 µg streptomycin (S)/mL (Gibco). Cells were transiently transfected with plasmid DNA using polyethylenimine (PEI) MAX, with a 1:4 (DNA:PEI MAX) ratio diluted in DMEM without serum and antibiotics. Cells were harvested 48 h post-transfection. Transient gene knockdowns of SMEK1 and SMEK2 were obtained through transient transfection of freshly seeded HEK293T cells, in DMEM with FCS but without P/S, with JetPrime (Polyplus) and the desired ON-TARGET plus siRNA SMARTpool mix from Horizon discovery (see S1

Table). Cells were transfected with 35 nM siRNA according to the JetPrime transfection protocol for siRNA. Cells transiently transfected with ON-TARGETplus non-targeting control pool served as a control. Optimal gene knockdown was achieved 72 h post-infection.

## Immunoblotting and immunoprecipitation

Total cell lysates were prepared by harvesting the cells and resuspending them in 10 pellet volumes of total cell lysis (TCL) buffer containing 50 mM Tris at pH 7.4, 150 mM NaCl, 0.1% sodium dodecyl sulfate (SDS), 0.5% sodium deoxycholate, 1% nonident P-40 (NP-40), 1 mM dithiothreitol (DTT), 1 mM PMSF, 1 mM benzamidine, 0.5 µg/mL LPP, 50 mM NaF, 10 mM β-glycerophosphate, 1 mM $Na_3VO_4$. Crude cell lysates were sonicated with a Hielscher handheld probe sonicator (5 sec., cycle 1, amplitude 70%) and spun down to pellet debris (15 000 x g, 10 min., 4°C). Cleared lysates were boiled in SDS sample buffer and analyzed by SDS-PAGE using Bis-Tris NuPAGE 4–12% gels. Proteins were blotted on a nitrocellulose membrane for 2 hours at 40 V in buffer containing 50 mM Tris and 50 mM boric acid, followed by Ponceau S staining to assess equal loading in all lanes. Next, the membranes were incubated overnight with primary antibody diluted in 5% milk/TBS-Tween20. Next, the membranes were washed three times with TBS-T, followed by 1 h incubation with HRP-coupled secondary antibody at room temperature.

Immunoblots were eventually analysed using an ImageQuant LAS 4000 and ECL reagent (PerkinElmer). Lysates for immunoprecipitations were prepared from cells from two 15 cm dishes, lysed in Co-IP buffer composed of 10 mM HEPES at pH 7, 150 mM NaCl, 6 mM $MgCl_2$, 0.5% NP-40, 10% glycerol, 2 mM DTT, 1 mM PMSF, 1 mM benzamidine, 0.5 µg/mL LPP. Crude lysates were sonicated with a Diagenode bioruptor sonicator (10 min., 15 sec. ON, 15 sec. OFF), followed by centrifugation (1000 x g, 10 min., 4°C). 60 µL of the cleared lysates was kept as input samples and boiled with sample buffer. The remaining lysate was either incubated with 30 µL of home-made anti-EGFP nanobodies covalently coupled with agarose beads for GFP-traps, or 30 µL anti-FLAG M2 affinity gel resin for FLAG-IPs. Lysates were incubated with the beads for 2 h at 4°C, before being washed with lysis buffer. The anti-EGFP nanobody beads were directly boiled in sample buffer, while the anti-FLAG M2 affinity gel resin was incubated with 3X-FLAG peptide for 1 h at 4°C to elute the proteins. Eluted proteins were then boiled in sample buffer.

## Split-luciferase assays

For the split-luciferase assays, protein-protein interaction sensors were made based on the NanoBiT technology (Promega) and the rationale described in Claes and Bollen, 2023 [24,27]. AlphaFold models shown in Figs 2E and 4A were used to guide the design of the sensors. SMEK1 was N-terminally tagged with the LgBiT-tag, by cloning SMEK1 cDNA into a vector containing the LgBiT-tag. Other proteins were fused with the SmBiT-tag. Flexible linkers, ranging from 10-15 amino acids, were included between the tags and the protein. Furthermore, all fusion proteins contained a FLAG-tag (Fig 1C), which allowed detection and comparison of expression levels by immunoblotting.

7 µg of the plasmids expressing the LgBiT- and SmBiT-tagged proteins were transiently transfected in a 15 cm dish containing HEK293T cells at a confluency of 25% using PEI Max as the transfection agent. Medium was changed for fresh medium 24 h post-transfection. 48 h post-transfection, cells were harvested and cell pellets were resuspended in 10 pellet volumes of assay buffer containing 50 mM Tris at pH 7.4, 150 mM NaCl, 10% glycerol, 0.01% saponin, 0.5 mM EDTA, 1 mM PMSF, 1 mM benzamidine, 0.5 µg/mL LPP and 1 mM DTT. Cell suspensions were freeze-thawed once, after which the lysates were cleared (20 000 x g, 5 min., 4°C). These cell lysates were further used for the lysate-based split-luciferase assays, as previous described [24,27].

In general, lysate-based split-luciferase assays were carried out in white, non-binding 384-well plates (Greiner) (deep-well or high-base). Cell lysates containing the SmBiT-tagged proteins were pipetted in the wells before cell lysate containing the LgBiT-tagged protein supplemented with the furimazine substrate (50 µM final) was added. For end-point measurements, the plate was analysed after 15–20 minutes incubation at room temperature, while for kinetic measurements, the plate was

read out every 10–20 seconds, until equilibrium was reached. All measurements were carried out at room temperature. For each measurement, a condition with the LgBiT-tagged cell lysate alone (addition of assay buffer instead of SmBiT-tagged cell lysate) was added. This accounted for the background signal, necessary to calculate the signal-to-background (S/B) ratio. Bioluminescence signal was measured using a Spark multimode microplate reader (Tecan Life Sciences). For competition assays, both LgBiT- and SmBiT-tagged cell lysates were mixed in a 384-well plate, after which the signal was continuously monitored every 10–20 seconds, until an equilibrium was reached. At this point, untagged competitor protein or assay buffer as a control was added to the wells, and the effect on the signal was monitored in function of time.

For split-luciferase assays with purified interaction sensors, 500 pM LgBiT-SMEK1 was mixed with 10 nM SmBiT-KAP1 or Rep68-SmBiT, together with 50 µM furimazine. Experimental procedures for end-point and kinetic measurements are the same as described above.

### Proximity ligation assay (PLA)

HeLa cells were seeded in a 12-well plate onto a poly-L-lysine coated coverslip at a confluency of 30%. The next day, part of the cells were co-infected with AAV2 (1000 IU) and Ad5 (MOI 5) to obtain expression of the Rep proteins in the context of a viral infection. 28h p.i. cells were processed for PLA staining according to the manufactures protocol, using the Duolink *In Situ* Orange Mouse/Rabbit kit (Sigma-Aldrich). Briefly, cells were washed three times with ice-cold PBS before being fixed with ice-cold methanol (100%) for 5 minutes at 4°C. Next, cells were washed again three times with PBS at RT and permeabilized for 10 minutes at RT in PBS supplemented with 0.5% triton X-100. The permeabilization was stopped by washing the cells three times with PBS-T (0.1% Tween20). From this point onwards, the manufacturer's instructions were followed. Primary antibodies for SMEK1 (Abcam ab70635, 1/150), KAP1 (Millipore MAB3662, 1/400) and Rep (Progen 65172, 1/10) were diluted in the provided Duolink Antibody diluent and incubated overnight at 4°C with the appropriate coverslip. Coverslips were incubated with 1 µg/mL DAPI for 10 minutes at RT, before being mounted on a glass slide using Fluoromount-G mounting medium. Finally, coverslips were imaged with a Leica TCSSPE laser-scanning confocal system mounted on a Leica DMI 4000B microscope, equipped with a Leica ACS APO 63X 1.30NA oil DIC objective.

### Microscale thermophoresis (MST)

MST assays were performed with GFP-tagged His-SMEK1 (His-SMEK1-GFP), His-Rep68 and His-KAP1 purified from *E. coli*. Proteins were purified as described above and dialysed to buffer containing 20 mM Tris (pH 7.5) and 150 mM NaCl. The buffer of the unlabelled ligands (His-Rep68 and His-KAP1) was exchanged right before use on an Amicon ultra centrifugal filter 30 kDa MWCO (Millipore) for fresh assay buffer (20 mM Tris pH 7.5, 150 mM NaCl) freshly supplemented with 0.05% Tween20 and 1 mM TCEP. After the final buffer exchange step, the proteins were maximally concentrated. All proteins were spun (1500 x g, 1 min.) through a Millex PVDF syringe filter column (Millipore) to remove aggregates that can interfere with the MST read-out. The fluorescently labelled target His-SMEK1-GFP was diluted in the same Tween20 and TCEP containing assay buffer, to reach a final concentration of 450 nM. Unlabelled ligands were diluted in the same buffer to create a serial dilution of 12 steps. Labelled- and unlabelled proteins were mixed (1:1 ratio) in a white, non-binding 384-well plate (Greiner, deep-well) and loaded onto a Monolith NT.Automated Premium Capillary Chip. Binding data was acquired using the Monolith NT.Automated (NanoTemper), with green laser intensity and MST power set at 40% and high, respectively. Data of three independent measurements were analysed in the MO.Affinty Analysis Software and data points were plotted as the bound fraction.

### Viral infections

HEK293T cells at a confluency of approximately 50% were infected with 10 infection units (IU) per cell of AAV2 and/or Ad5, at a multiplicity of infection (MOI) 5. The volume of culture medium was reduced to 2/5 of the original well volume

before addition of AAV2. Cells with AAV2 were incubated for 2 h at 37°C before Ad5 was added. 1 h after Ad5 addition, the original cell culture medium was restored by addition of complete DMEM medium. The cells were harvested 28 h post-infection.

Where necessary, cells were pre-treated with DMSO or 3 µM KU-60019, a selective ATM inhibitor, 4 h prior to (r)AAV infection. The inhibitor was maintained throughout the experiment.

## AAV-based replication assays

HEK293T cells harvested 28 h post-infection with AAV2 and/or Ad5 were divided over two clean reaction tubes. Cells were pelleted and one tube was stored at -80°C for genomic (g)DNA extraction. The other cell pellet was lysed in total cell lysis (TCL) buffer containing 50 mM Tris at pH 7.4, 150 mM NaCl, 0.1% sodium dodecyl sulfate (SDS), 0.5% sodium deoxycholate, 1% nonident P-40 (NP-40), 1 mM dithiothreitol (DTT), 1 mM PMSF, 1 mM benzamidine, 0.5 µg/mL LPP, 50 mM NaF, 10 mM β-glycerophosphate, 1 mM $Na_3VO_4$. Crude lysates were sonicated with a Hielscher handheld probe sonicator (5 sec., cycle 1, amplitude 70%) and cleared at maximum speed via centrifugation (10 min., 21 000 x g, 4°C). Lysates mixed with SDS sample buffer were loaded on a SDS PAGE gel and blotted as described above. Blots were analyzed for Rep and VP expression levels, as well as the phosphorylation state of KAP1[S824].

gDNA and AAV2 viral genomes were extracted from the second cell pellet using the GenElute Mammalian Genomic DNA miniprep kit (Sigma-Aldrich). qPCR was performed to quantify the viral genomes per cell (vg/cell). A standard curve of HEK293T gDNA and cut pAV2 plasmid was included to allow the precise quantification of the number of cells and AAV2 genomes [5]. For each sample, two different primer pairs for two different genes (cyclophilin and AAV2 *cap*) were used. All vg/cell values were normalized to the control group used in each experiment, either NT siRNA or shRNA Luc cell line. For primer pair sequences, see S1 Table.

## AAV2 ELISA

Cells harvested for AAV2 viral particle (VP) quantification were lysed in buffer containing 50 mM Tris (pH 8), 150 mM NaCl and 2 mM MgCl2. Crude cell lysates were subjected to three freeze-thaw cycles before being incubated with 10 U benzonase endonuclease for 1h at 37°C. The debris was spun down (1500 x g, 10 min., 4°C) and the total protein concentration was determined via the bicinchoninic acid (BCA) assay following the manufacturer's protocol (Thermo Scientific). Next, the number of viral particles per µg total protein extract was determined using the AAV2 titration ELISA kit (Progen). Lysates were further diluted to a final concentration of 3.3 ng/µL in ASSB (1X) assay buffer provided in the ELISA kit, before performing the ELISA according to the manufacturer's instructions. Samples were measured in duplicate, and the number of viral particles for each sample was determined according to the included AAV2 standard curve. The VP/µg total protein extract of each sample was normalized to the shRNA Luc cell line with Dox treatment.

## AlphaFold 3 multimer prediction

For AlphaFold 3 multimer predictions, the AlphaFold server from google DeepMind was used (https://alphafoldserver.com). Primary amino acid sequences of the proteins were obtained from Uniprot. Protein sequences were submitted on the AlphaFold server to model the interaction between the proteins. Models were analysed via UCSF ChimeraX.

## Supporting information

**S1 Methods. Generation of shRNA knockdown cell lines and thermal shift assays.**
(DOCX)

**S1 Table. Used materials.**
(XLSX)

**S1 Fig. SMEK1 interacts with KAP1 and the four Rep isoforms.** (A) Quantification of the immunoblot GFP-SMEK1, Rep78 and KAP1 band intensities both for the Input and GFP-trap conditions. Rep78/GFP-SMEK1 or KAP1/GFP-SMEK1 ratios of the GFP-trap conditions were divided by the same ratio of the corresponding input condition. Quantification was done for two independent experiments. (B) Proximity ligation assay (PLA) for the interaction between SMEK1:KAP1 in non-infected HeLa cells and for the SMEK1:Rep interaction in AAV2 (1000 IU) and Ad5 (MOI 5) co-infected HeLa cells. Cells incubated with either the single anti-SMEK1, -KAP1 or -Rep antibody, or no primary antibodies, served as a negative control. All conditions were incubated with the PLA probes. Orange dots indicate the association between SMEK1 and either KAP1 or Rep. Nuclei were counterstained with DAPI. Scale bar = 25 µm. (C) GFP-Trap experiment of GFP-tagged SMEK1 from cells ectopically expressing FLAG-tagged Rep (right panel). GFP expression alone served as a control (left panel). Input samples are shown on the left, while trap samples are shown on the right.
(TIF)

**S2 Fig. Validation of the SMEK1:Rep interaction.** (A) Design of the split-luciferase sensors and untagged competitors used for the protein-protein interaction studies with purified proteins. Sensors and competitors were tagged N-terminally with a 10X His-tag for recombinant expression and purification from *E. coli*. (B) Coomassie staining and immunoblot visualization of the purified split-luciferase sensors and untagged competitors represented in S2A Fig and used in Figs 1E and S2C. The star indicates the respective species. (C) Kinetic-trace experiment with the LgBiT-SMEK1:Rep68-SmBiT and LgBiT-SMEK1:SmBiT-KAP1 purified interaction sensors. 0.5 nM LgBiT-SMEK1 was mixed with 10 nM of SmBiT-tagged protein. The black arrow indicates the addition of untagged competitor. Concentration of the competitors is indicated in the graph. The presented data is plotted as a percentage of the signal-to-background (S/B) ratio right before the addition of competitor. (D) Coomassie staining of FLAG-SMEK1 and FLAG-SMEK2 ectopically expressed in and purified from HEK293T cells. (E) Coomassie staining of the purified proteins used in the microscale thermophoresis (MST) assays shown in Fig 1G.
(TIF)

**S3 Fig. The HEAT/Arm domain of SMEK1 is essential for its interaction with Rep68.** (A) Design of the truncated FLAG-LgBiT-SMEK1 split-luciferase sensors. Sensors were made for expression in HEK293T cells or *E. coli* (His-tagged). (B) Split-luciferase assays with the purified truncated LgBiT-SMEK1 (0.5 nM) and Rep68-SmBiT (10 nM) interaction sensors. End-point measurements were taken after mixing the sensor components and incubating them at room temperature for 20 minutes prior to read-out. Statistical significance was determined by an one-way ANOVA with Dunnett's multiple comparison test. (C) Coomassie staining of the His-LgBiT-SMEK1 deletion mutants purified from *E. coli*.
(TIF)

**S4 Fig. The Rep68:SMEK1 interaction is independent of PP4 association.** (A) Immunoblots of the split-luciferase lysates used in Fig 2F. (B) Thermal shift assay of WT and R557→A/E LgBiT-SMEK1 in lysates to assess the effect on the thermal stability of SMEK1 upon introduction of a point mutation in the HEAT/Arm domain. (C) FLAG-IP results of ectopically expressed FLAG-SMEK1 (WT and R557 mutants) showing loss of binding between endogenous PP4 and FLAG-SMEK1$^{R557→A/E}$, while the interaction with ectopically expressed Rep68 and endogenously expressed SF3B1 is unaffected. (D) Immunoblot of the split-luciferase lysates used in Fig 2G.
(TIF)

**S5 Fig. KAP1$^{S824}$ phosphorylation is independent of ATM activation upon (r)AAV2 infection.** (A) HEK293T cells were pre-treated with a selective ATM inhibitor (KU-60019) ($C_{fin}$ = 3 µM) prior to infection/transduction with increasing (r)AAV2 IUs and a fixed Ad5 MOI of 5. pKAP1$^{S824}$ levels were assessed via immunoblotting 28h post-infection/transduction. (B) Quantification of the PP4 protein levels in the four main groups of the experiment shown in Fig 3B (mean ±SD; n = 12). Statistical significance was determined by an one-way ANOVA with Dunnett's multiple comparison test. (C) Quantification

of the SMEK1/2 protein levels in the five main groups of the experiment shown Fig 3D (mean ±SD; n = 18). Statistical significance was determined by an one-way ANOVA with Dunnett's multiple comparison test.
(TIF)

**S6 Fig. SMEK1 and SMEK2 exhibit extensive sequence and structural homology.** (A) Primary amino acid sequence alignment of SMEK1 and SMEK2, with only differing residues highlighted. Hydrophobic, negatively charged, positively charged, and other amino acids are color-coded in blue, orange, red, and grey, respectively. Sequence alignment was created with Jalview. (B) AlphaFold 3 multimer predictions of the PP4:PP4R2:SMEK1 and PP4:PP4R2:SMEK2 heterotrimeric complexes, showing the structural homology between the two complexes. (C) DepMap CRISPR knockout data showing a functional dependency between SMEK1 and SMEK2. (D) GFP-trap of GFP-tagged SMEK1 and SMEK2 from cells co-expressing Rep68.
(TIF)

**S7 Fig. The [423]MAPP[426] SLiM motif of KAP1 interacts with SMEK1 and SMEK2.** (A) Immunoblot of the LgBiT-SMEK1[WT/ΔEVH1] and SmBiT-KAP1[WT/MAPP] split-luciferase lysates used in Figs 4B and S7B. (B) Lysate-based split-luciferase end-point measurement of the LgBiT-SMEK1[WT]:SmBiT-KAP1 and LgBiT-SMEK1[ΔEVH1]:SmBiT-KAP1 interaction sensors. Bioluminescence signal was read out after 25 minutes incubation at room temperature and plotted as a percentage of the LgBiT-SMEK1[WT]:SmBiT-KAP1 signal (mean ± SD; n = 3 independent repeats). Statistical significance was determined by a two-tailed unpaired t-test. (C) GFP-trap of ectopically expressed GFP-SMEK1 (WT and ΔEVH1) assessing the co-precipitation of ectopically expressed FLAG-KAP1 (WT and MAPP mutant). (D) Structural alignment of the EVH1:FKVP co-crystal structure (PDB 6R8I) with the AlphaFold 3 model 1 prediction shown in Fig 4C. (E) Coomassie staining of the purified YFP-4xFKVP and YFP-4xAKVA fusion proteins performed to assess the purity. (F) Kinetic-trace experiment of the LgBiT-SMEK2:SmBiT-KAP1 interaction sensor. Arrow indicates the addition of purified YFP-4xFKVP competitor or the AKVA control (concentrations indicated in the graph). The represented data is plotted as a percentage of the signal-to-background (S/B) ratio right before addition of the competitor.
(TIF)

**S8 Fig. Rep68 interacts with SMEK2 and is outcompeted by the FKVP peptide.** (A) Kinetic-trace experiment of the LgBiT-SMEK2:Rep68-SmBiT interaction sensor. Arrow indicates the addition of purified YFP-4xFKVP competitor or the AKVA control (concentrations indicated in the graph). The represented data is plotted as a percentage of the S/B ratio right before addition of the competitor. (B) Kinetic-trace experiment showing the time-dependent association of the LgBiT-B56$_\delta$:SmBiT-RepoMan split-luciferase sensor. The black arrow indicates the addition of the purified YFP-competitor peptides (concentration indicated in the graph). The represented data is plotted as a percentage of the S/B ratio right before the addition of the competitor. The data shown is a representative example of three independent repeats.
(TIF)

**S9 Fig. The HEAT/arm and/or C-term of SMEK1 interact(s) with NIPP1.** GFP-trap of GFP-tagged SMEK1 (WT, ΔEVH1, ΔHEAT+C-term) or GFP alone (control) to check for co-precipitation of endogenous NIPP1.
(TIF)

## Acknowledgments

We thank Benjamien Moeyaert, Samir Nuseibeh, and Sofie De Munter for their valuable feedback on the written manuscript. We would like to thank Valentina Zorzini of the Biophysics Expertise Unit, VIB-KU Leuven Center for Brain & Disease Research, 3000 Leuven, Belgium for her help with the MST assays. All figures and schematic representations were created with Adobe Illustrator.

## Author contributions

**Conceptualization:** Bram Vandewinkel.

**Formal analysis:** Bram Vandewinkel.

**Funding acquisition:** Bram Vandewinkel, Mathieu Bollen, Els Henckaerts.

**Investigation:** Bram Vandewinkel, Sophie Torrekens, Zander Claes.

**Supervision:** Bram Vandewinkel, Mathieu Bollen, Els Henckaerts.

**Validation:** Bram Vandewinkel.

**Visualization:** Bram Vandewinkel.

**Writing – original draft:** Bram Vandewinkel.

**Writing – review & editing:** Bram Vandewinkel, Zander Claes, Mathieu Bollen, Els Henckaerts.

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
