## [Decision Letter · Decision Letter 0]

22 Nov 2025

The adeno-associated virus Rep proteins target PP4:SMEK1 by preventing substrate-recruitment

PLOS Pathogens

Dear Dr. Vandewinkel,

Thank you for submitting your manuscript to PLOS Pathogens. After careful consideration, we feel that it has merit but does not fully meet PLOS Pathogens's publication criteria as it currently stands. Therefore, we invite you to submit a revised version of the manuscript that addresses the points raised during the review process. Please address the concerns raised by all three reviewers about quantifications, statistical analysis and additional control experiments. Specifically, please provide data measuring virus replication using genome copies and particle production, details on a controlled inducible PP4 knockdown experiment, and specifics on how Rep-SMEK1 interaction occurs (R1); perform silencing studies under conditions where the knockdowns are more complete so that the findings more biologically significant (R2); and improve the rigor of lysate-based split-luciferase system with additional critical controls (R3). I also encourage you to address R2’s suggestion for SPR studies using experiments or modifications to the relevant section of the results and discussion.

We look forward to receiving your revised manuscript.

Kind regards,

Kinjal Majumder, PhD

Guest Editor

PLOS Pathogens

Donna Neumann

Section Editor

Editor-in-Chief

PLOS Pathogens

Editor-in-Chief

PLOS Pathogens

orcid.org/0000-0002-7699-2064

**Journal Requirements:**

At this stage, the following Authors/Authors require contributions: Bram Vandewinkel, Sophie Torrekens, Zander Claes, Mathieu Bollen, and Els Henckaerts. Please ensure that the full contributions of each author are acknowledged in the "Add/Edit/Remove Authors" section of our submission form.

https://journals.plos.org/plospathogens/s/submission-guidelines#loc-parts-of-a-submission

4) We do not publish any copyright or trademark symbols that usually accompany proprietary names, eg ©,  ®, or TM  (e.g. next to drug or reagent names). Therefore please remove all instances of trademark/copyright symbols throughout the text, including:

- ® on page: 21

- TM on page: 19.

5) Please upload all main figures as separate Figure files in .tif or .eps format. For more information about how to convert and format your figure files please see our guidelines:

6) We have noticed that you have uploaded Supporting Information files, but you have not included a list of legends. Please add a full list of legends for your Supporting Information files after the references list.

7) Some material included in your submission may be copyrighted. According to PLOSu2019s copyright policy, authors who use figures or other material (e.g., graphics, clipart, maps) from another author or copyright holder must demonstrate or obtain permission to publish this material under the Creative Commons Attribution 4.0 International (CC BY 4.0) License used by PLOS journals. Please closely review the details of PLOSu2019s copyright requirements here: PLOS Licenses and Copyright. If you need to request permissions from a copyright holder, you may use PLOS's Copyright Content Permission form.

Potential Copyright Issues:

i) Figures 1A, 1D, 4E, 5, 7, and S7B. Please confirm whether you drew the images / clip-art within the figure panels by hand. If you did not draw the images, please provide (a) a link to the source of the images or icons and their license / terms of use; or (b) written permission from the copyright holder to publish the images or icons under our CC BY 4.0 license. Alternatively, you may replace the images with open source alternatives. See these open source resources you may use to replace images / clip-art:

8) Please amend your detailed Financial Disclosure statement. This is published with the article. It must therefore be completed in full sentences and contain the exact wording you wish to be published.

9)  Please ensure that the funders and grant numbers match between the Financial Disclosure field and the Funding Information tab in your submission form. Note that the funders must be provided in the same order in both places as well.

10) Kindly revise your competing statement in the online submission form to align with the journal's style guidelines: 'The authors declare that there are no competing interests.'

**Reviewers' Comments:**

Reviewer's Responses to Questions

**Part I - Summary**

Reviewer #1: This study identifies the PP4:SMEK1/2 phosphatase complex as a key regulator of wild-type adeno-associated virus (AAV) replication. The authors show that AAV Rep68 binds SMEK1 to inhibit PP4 activity, leading to hyperphosphorylation of KAP1 and RPA2, which enhances viral gene expression and replication. They further describe how interactions between KAP1, SMEK1, and associated phosphatase complexes control this process, revealing a novel mechanism by which viruses modulate host phosphatases to promote viral replication.

The work is interesting and reveals a new mechanism in AAV-helpervirus-host cell interaction, but the study could be strengthened by adding important experimental controls.

Reviewer #2: The manuscript presents a compelling biochemical model in which AAV Rep proteins directly interact with the PP4:SMEK1/2 phosphatase holoenzyme to inhibit substrate recruitment, thereby sustaining hyperphosphorylation of KAP1 and RPA2 to promote AAV replication. However, the functional genetic data provided to support this model, specifically the PP4 and SMEK1/2 knockdown experiments in Figures 3 are not sufficiently robust to substantiate the claimed biological significance.

Reviewer #3: The authors describe here the interactions of AAV Rep proteins with PP4:SMEK1/2 and show that binding of Rep to this phosphatase blocks interaction and dephosphorylation of a known substrate, KAP1 which results in activation of the lytic cycle. They use a combination of assays to study the interaction, together with AlphaFold3 model prediction to guide the design of their mutants. Overall, this is a very interesting paper that show how viral proteins modulate phosphatase activity. However, my major concern is the lack of controls in expression levels when performing the split-luciferase experiments using cellular extracts.

**Part II – Major Issues: Key Experiments Required for Acceptance**

Reviewer #1: 1. The authors use AAV2 Rep and Cap protein accumulation as a surrogate for viral replication. This should be validated at the level of viral DNA synthesis (e.g., Southern blot or qPCR of AAV genomes) and at the level of particle production (titration). Please provide these measurements (with replicates) or justify why the current readout is sufficient.

2. Line 154–155: state explicitly how the apparent reduction of the SMEK1–Rep interaction after crosslinking was quantified (e.g., densitometry of specific bands normalized to input, number of biological replicates, statistical test). If quantification was performed, include the raw data and statistical summary.

3. For the PP4 knockdown experiments, a cell line with doxycycline-inducible expression of a scrambled shRNA is a more appropriate control than an empty vector. Also, the experiment should be performed both in the presence and absence of doxycycline and knockdown efficiency and reproducibility (biological replicates) reported.

Reviewer #2: 1. In Fig. 1F, the authors show co-precipitation of purified Rep68 and KAP1 with purified FLAG-SMEK1 (and SMEK2) using bead-based pulldown assays. However, these experiments are strictly qualitative and do not allow conclusions regarding binding affinity, stoichiometry, or specificity beyond binary association. Given that this interaction is a central mechanistic claim that AAV Rep68 directly binds SMEK1/2 and competes with cellular substrates, the manuscript would be significantly strengthened by quantitative biophysical measurements, such as SPR.

2. The knockdown efficiencies for PP4C, SMEK1, and SMEK2 are clearly incomplete. As a result, the observed changes in Rep expression and AAV genome replication are modest (~2‑fold or less). In particular, the only two-fold in virus genome replication does not consider biologically significant. Furthermore, several comparisons show poor statistical significance (high P values). Given that the central model proposes that PP4:SMEK1/2 strongly suppresses AAV replication and that Rep68 exerts a dominant inhibitory effect on PP4, the current functional data do not demonstrate a robust biological requirement for PP4 or SMEK1/2.

Reviewer #3: In all lysate-based split-luciferase assays the assumption is that all mutants/deletion mutants are expressed to the same level. However, this is not tested (or not shown). E.g. in Figure 2B the observed increase in interaction between SMEK1/deltaC or deltaEVH1+C with PP4 could this because this deletion mutant is expressed at much higher levels and therefore it appears that there is an increased interaction? Likewise, how well are the other deletion mutants express where you see a complete loss of binding? Expression of the LgBiT and SmBiT fusion constructs in panels E and F should also be shown to ensure that any observed phenotype is indeed due to a change in interaction/affinity rather than a change in expression levels. And for the mutant expressed in panel 4B- expression of both WT and mutant should be verified.

**Part III – Minor Issues: Editorial and Data Presentation Modifications**

Reviewer #1: 1. Indicate the position of the FLAG-tags of the SmBit- and LgBit- constructs in the schematic illustrations.

2. Explain the rationale for placing SmBit/LgBit at the chosen N- or C-termini of the tested proteins. If orientation could affect folding or interaction, provide data showing that tag position does not alter protein expression, localization, or binding (or include this as a caveat).

3. The Coomassie stain shown in Fig. S1C should be corroborated by immunoblotting for the relevant species to confirm band identity and loading.

4. Line 230: for clarity please revise the sentence to read: “The PP4-binding mutants of SMEK1 still interacted …”

5. Line 232: there is an inconsistency between the main text (“GFP-SMEK1”) and Fig. S3C (“FLAG-SMEK1”).

6. For pull-down experiments where GFP is used as a control for GFP-SMEK fusions (e.g., Fig. 1B, Fig. 6B/C), using GFP fused to an unrelated protein of similar size would be a more appropriate control than GFP alone.

7. The results shown in Fig. 4B should be validated by an independent biochemical approach (reciprocal pull-down or co-immunoprecipitation), with quantification of the interaction across replicates and appropriate negative controls.

Reviewer #2: Quantification of the Western Blots in Fig. 3F is needed. Figure 3F presents qualitative blots showing increased phosphorylation of RPA2 and KAP1 with increasing AAV input. These data are consistent with the proposed model, but no quantification is provided.

Reviewer #3: 1. In Figure 1B: does the Rep antibody used detect all 4 Rep proteins? Please give additional information in the figure legend.

2. Lines 153-156: “The enzyme-substrate interaction between SMEK1 and KAP1 was further enhanced by prior crosslinking, whereas the SMEK1:Rep interaction diminished upon crosslinking, possibly caused by stabilization of more dynamic SMEK1 protein complexes (Fig. 1B).” There are other possibilities why less protein is recovered when using crosslinkers, e.g. the epitope might be partly masked due to the crosslinking. Please, include as an alternative.

3. Figure 1F: How does “purified” Flag-SMEK1 or 2 look like without PD. Can you detect them in Coomassie stained gel – how “clean” are they? Please include lanes where only Flag-SMEK1/2 are loaded.

4. Fig. 3F – in the text (line 279) it is mentioned that increasing AAV in the presence of a fixed MOI of Ad5 is used, but in the figure it looks like both AAV and Ad5 are used in increasing amounts – please correct.

5. Line 623, typo, AAV2 rather than AA2.

6. Figure legend. Figure 1B – Rep can’t be endogenous, it is a viral protein and therefore by definition is exogenous – please correct.

7. Figure 3A – wild-type and recombinant AAV2 is used but nowhere in the text is explained what the difference is between the two. Please add this in the text.

PLOS authors have the option to publish the peer review history of their article (what does this mean? ). If published, this will include your full peer review and any attached files.

**Do you want your identity to be public for this peer review?** For information about this choice, including consent withdrawal, please see our Privacy Policy .

Reviewer #1: No

Reviewer #2: No

Reviewer #3: No

**Figure resubmission:**

**Reproducibility:**



---

## [Editor Report · Decision Letter 1]

23 Feb 2026

Dear Mr. Vandewinkel,

We are pleased to inform you that your manuscript 'The adeno-associated virus Rep proteins target PP4:SMEK1 by preventing substrate-recruitment' has been provisionally accepted for publication in PLOS Pathogens.

Best regards,

Kinjal Majumder, PhD

Guest Editor

PLOS Pathogens

Donna Neumann

Section Editor

PLOS Pathogens

Sumita Bhaduri-McIntosh

Editor-in-Chief

PLOS Pathogens

orcid.org/0000-0003-2946-9497

Michael Malim

Editor-in-Chief

PLOS Pathogens

orcid.org/0000-0002-7699-2064
---

## [Editor Report · Acceptance letter]

Dear Mr. Vandewinkel,

We are delighted to inform you that your manuscript, "The adeno-associated virus Rep proteins target PP4:SMEK1 by preventing substrate recruitment," has been formally accepted for publication in PLOS Pathogens.

Best regards,

Sumita Bhaduri-McIntosh

Editor-in-Chief

PLOS Pathogens

orcid.org/0000-0003-2946-9497

Michael Malim

Editor-in-Chief

PLOS Pathogens

orcid.org/0000-0002-7699-2064